# Bench2Drive: Towards Multi-Ability Benchmarking of Closed-Loop End-To-End Autonomous Driving

**Xiaosong Jia***     **Zhenjie Yang***     **Qifeng Li***     **Zhiyuan Zhang***

**Junchi Yan**[†]
[*] Equal contributions.     [†] Correspondence author

Dept. of CSE & School of AI & MoE Key Lab of AI, Shanghai Jiao Tong University

https://thinklab-sjtu.github.io/Bench2Drive/

## Abstract

In an era marked by the rapid scaling of foundation models, autonomous driving technologies are approaching a transformative threshold where end-to-end autonomous driving (E2E-AD) emerges due to its potential of scaling up in the data-driven manner. However, existing E2E-AD methods are mostly evaluated under the open-loop log-replay manner with L2 errors and collision rate as metrics (e.g., in nuScenes), which could not fully reflect the driving performance of algorithms as recently acknowledged in the community. For those E2E-AD methods evaluated under the closed-loop protocol, they are tested in fixed routes (e.g., Town05Long and Longest6 in CARLA) with the driving score as metrics, which is known for high variance due to the unsmoothed metric function and large randomness in the long route. Besides, these methods usually collect their own data for training, which makes algorithm-level fair comparison infeasible.

To fulfill the paramount need of comprehensive, realistic, and fair testing environments for Full Self-Driving (FSD), we present **Bench2Drive**, the first benchmark for evaluating E2E-AD systems' multiple abilities in a closed-loop manner. Bench2Drive's official training data consists of 2 million fully annotated frames, collected from 13638 short clips uniformly distributed under 44 interactive scenarios (cut-in, overtaking, detour, etc), 23 weathers (sunny, foggy, rainy, etc), and 12 towns (urban, village, university, etc) in CARLA v2. Its evaluation protocol requires E2E-AD models to pass 44 interactive scenarios under different locations and weathers which sums up to 220 routes and thus provides a comprehensive and disentangled assessment about their driving capability under different situations. We implement state-of-the-art E2E-AD models and evaluate them in Bench2Drive, providing insights regarding current status and future directions.

## 1  Introduction

In recent years, the field of autonomous driving has witnessed tremendous growth, fueled by the rapid advancement and scaling of foundation models [1–3]. These developments have ushered in a new era of end-to-end autonomous driving (E2E-AD) systems [4–8], which promise a scalable, data-driven approach to vehicle automation, opposed to traditional module-based perception [9–13],

This work was in part supported by NSFC (92370201, 62222607) and Shanghai Municipal Science and Technology Major Project under Grant 2021SHZDZX0102.

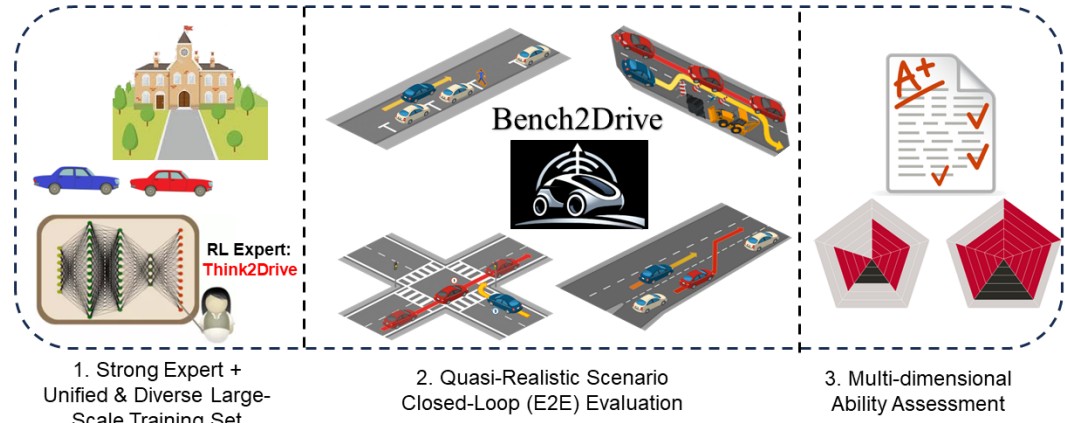

Figure 1: **Overview of Bench2Drive**.

prediction [14–17], planning [18–20] pipeline. Such systems are designed to be capable of learning from vast amounts of data, potentially transforming the landscape of vehicle intelligence.

Despite these advancements, **the evaluation methodologies for E2E-AD systems remain a critical bottleneck**. One popular way is to conduct log-replay with the recorded expert trajectories in dataset like nuScenes [21], i.e., open-loop evaluation. These models [4, 22] usually predict the future locations of the ego vehicle with the raw sensor information as inputs. As for metrics, the L2 error relative to the recorded trajectories and the ratio of collision happening are used. However, as widely discussed in the community [23, 24, 19], these open-loop metrics are insufficient for showcasing proficiency in planning, due to issues including distribution shift [25], causal confusion [26, 27], etc. nuScenes is also problematic due to its small and imbalanced validation set (around 75% of the frames only require continuing to drive straight) [24]. As a result, only encoding the ego status (location, speed, etc) [23] could achieve similar L2 errors compared to complex methods with sensor inputs [4], which **prompts a call for a closed-loop evaluation benchmark for E2E-AD**.

CARLA [28] is one of the most widely used simulator for closed-loop E2E-AD evaluation. Within its framework, benchmarks such as Town05Long and Longest6 have been established, featuring multiple routes that require AD systems to complete safely within specific time constraints. However, these benchmarks only assess basic skills such as lane following, making turns, collision avoidance, and traffic lights obeying [29, 30], **failing to examine AD systems' driving ability under complicated and interactive traffic**. The latest CARLA Leaderboard v2 introduces 39 challenging scenarios designed to evaluate the robustness of AD systems in more intricate situations. Nevertheless, the official routes for evaluation, ranging from 7 to 10 kilometers and filled with scenarios, present a formidable challenge, often too difficult to complete flawlessly, as shown in Fig. 2 (a). Consequently, with the driving score metric employing an exponential decay function, it becomes challenging to effectively compare different AD systems, as they tend to score very low. For instance, in the current Leaderboard v2[1], participating methods score less than 10 points out of 100. Besides, existing methods usually collect data by themselves which makes algorithm-level fair comparison infeasible.

To address the aforementioned challenges in evaluating autonomous driving (AD) systems, it is essential to develop a new benchmark that fairly assesses their capabilities in a granular manner. To this end, we introduce **Bench2Drive**, a new benchmark designed to evaluate E2E-AD systems in a comprehensive, realistic, and fair closed-loop environment. Bench2Drive has **an official training dataset** collected by state-of-the-art expert model *Think2Drive* [31], comprising 2 million fully annotated frames, sourced from 13638 clips. It span a diverse array of 44 interactive scenarios such as cut-ins, overtakings, and detours under different weather conditions and towns, ranging from sunny days in bustling city centers to foggy conditions in quaint villages. **The evaluation protocol includes 220 short routes**, each only around 150 meters in length and containing a single specific scenario. In this way, the assessment of individual skills is isolated and thus allows for a detailed comparison of the AD systems' proficiency across 44 distinct skill sets. Moreover, the brevity of each route mitigates the impact of the exponential decay function on the driving score, facilitating a more accurate and meaningful comparison of performance across different systems. Such a structured and focused

---

[1]https://eval.ai/web/challenges/challenge-page/2098/leaderboard/4942

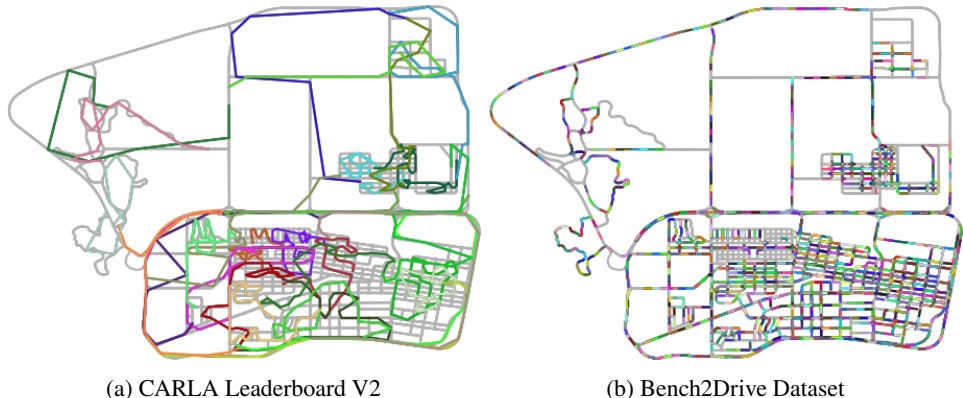

(a) CARLA Leaderboard V2           (b) Bench2Drive Dataset

Figure 2: **Route length on Town12.** We use different colors to represent different routes. Bench2Drive's short routes provide more smoothed evaluations.

benchmark would provide clearer insights into the strengths and weaknesses of each AD system, enabling targeted improvements and more refined technology development.

In summary, the proposed Bench2Drive benchmark features:

- **Comprehensive Scenario Coverage**: Bench2Drive is designed to test AD systems across 44 interactive scenarios, providing a thorough evaluation about capabilities under complex situations.
- **Granular Skill Assessment**: By structuring the evaluation across 220 short routes, each focusing on a specific driving scenario, Bench2Drive allows for detailed analysis and comparison of how different AD systems perform on individual tasks.
- **Closed-Loop Evaluation Protocol**: Bench2Drive evaluates AD systems in a closed-loop manner, where the AD system's actions directly influence the environment. This setup offers an accurate assessment of an AD system's driving performance.
- **Diverse Large-Scale Official Training Data**: Bench2Drive consists of a standardized training set of 2 million fully annotated frames from 13638 clips under diverse scenarios, weathers, and towns, ensuring that all AD systems are trained under abundant yet similar conditions, which is crucial for fair algorithm-level comparisons.

*These features make Bench2Drive a pioneering benchmark in the field of autonomous driving, providing an essential tool for researchers to refine and evaluate their E2E-AD systems in a realistic, comprehensive, and fair manner.* **We implement several classic baselines including TCP [5], ThinkTwice [30], DriveAdapter [27], UniAD [4], VAD [22], and AD-MLP [23] and evaluate them in the Bench2Drive**. We confirm the fact that open-loop metrics like L2 error could not reflect the actual driving performance. For the classic closed-loop metric - Drive Score, we find that it lack details and its heavy punishment encourages over-conservative driving strategies while Bench2Drive offers a comprehensive understanding about capabilities of different methods.

## 2 Related Work

### 2.1 Planning Benchmarks

Benchmarking in the field of autonomous driving has evolved from specialized datasets, such as KITTI [32] for perception and NGSIM/highD [33], BARK [34] for behavior prediction, to integrated forms like nuScenes [21], Argoverse [35], and Waymo [36], which facilitate the evaluation of various synergic system components. Recently, the assessment of planning capabilities for learning-based methods has become an area of interest [37–41]. In Table 1, we present a comparison of planning benchmarks. nuScenes [21], while offering open-loop metrics, has been critiqued for its inability to adequately evaluate planning proficiency due to the lack of closed-loop simulation [23, 24, 19]. Furthermore, it suffers from an imbalanced validation set, with a significant portion (75%) of scenarios only requiring straightforward driving, thus inadequately challenging the decision-making capabilities of AD systems in complex environments [24]. nuPlan [38] and Waymax [37] offer closed-loop evaluations but are limited to bounding box level assessments, excluding sensor simulation and, consequently, are not suitable for E2E-AD methods. Longest6 [6], a modified version of CARLA Leader-

Table 1: **Comparison with related planning benchmarks** Bench2Drive is the only benchmark to evaluate the E2E-AD methods under closed-loop with multi-ability analysis.

| Benchmark | Sensor | Closed-Loop | E2E-Sim | Expert | Complex | Multi-Ability-Eval |
|---|---|---|---|---|---|---|
| nuScenes [21] | ✓ | ✗ | ✗ | ✓ | ✗ | ✗ |
| nuPlan [38] | ✓ | ✓ | ✗ | ✓ | ✓ | ✗ |
| Waymax [37] | ✗ | ✓ | ✗ | ✓ | ✓ | ✗ |
| Longest6 [6] | ✓ | ✓ | ✓ | ✓ | ✗ | ✗ |
| CARLA LB V2 [28] | ✓ | ✓ | ✓ | ✗ | ✓ | ✗ |
| **Bench2Drive (Ours)** | ✓ | ✓ | ✓ | ✓ | ✓ | ✓ |

board V1, only assesses basic skills such as lane following, making turns, collision avoidance, and traffic lights. CARLA Leaderboard V2 [28] lacks expert demonstration data. As widely discussed in the community [42, 43], the lack of an official training set makes the comparisons of different methods in the system-level instead of the algorithm-level.Bench2Drive deal with these shortcomings by offering a large-scale, annotation-rich official training dataset alongside a multi-ability evaluation set. This enables a more granular and informative assessment of an AD system's driving capabilities, overcoming the limitations of existing benchmarks that rely on average scoring across all routes as their primary performance metric.

## 2.2 End-to-End Autonomous Driving

The concept of E2E-AD could date back to 1980s [44]. Recently, the arise of neural network, especially Transformer [45], demonstrates the power of scaling laws, which rejuvenates the enthusiasm for E2E-AD [46–50]. However, they are either evaluated only in the open-loop way [51, 4, 22, 52] or in the relatively simple scenes like Town05Long/Longest6 [53, 54, 42, 55–60]. Bench2Drive offers a challenging and comprehensive arena to compare E2E-AD methods' ability.

## 3 Bench2Drive

Bench2Drive consists of a large-scale fully annotated dataset collected in CARLA as the official training set, an evaluation toolkit for the granular driving skill assessment, and implementations of several state-of-the-art E2E-AD methods tailored for the training dataset and evaluation toolkit. All data, codes, and checkpoints are in GitHub and Huggingface under Apache License 2.0. We give details in the following section.

### 3.1 Data Collection Agent

The data collection agent (expert) is responsible for collecting the data so that student models could learn from the data. In the real world, this is usually done by human to drive around the city, like the curation of KITTI [32], nuScenes [21], Waymo [36], Argoverse [35]. However, it requires lots of human efforts. In simulation, there is a cheap substitute - teacher model. The teacher model would use information not available in the real world (termed privileged information), for example, ground-truth locations, states, and intentions of surrounding agents and ground-truth states of traffic lights, etc. As a result, people using CARLA either write rules [43, 61] or train a RL model [50, 31] to use the privileged information to drive in the simulation.

In this work, we use the world model based reinforcement learning teacher - Think2Drive [31] to navigate in CARLA and collect data, since **it is the only expert model which is able to solve all 44 scenarios during the construction of Bench2Drive**. Notably, after the release of Bench2Drive, the rule-based expert PDM-Lite [61][2] is open sourced and users could use it for customized demand.

### 3.2 Expert Dataset

Existing E2E-AD methods evaluated in the closed-loop manner [5, 6, 56, 57] typically collect their own data using the CARLA simulator. However, as highlighted in [42, 43], the size and distributions

---

[2]https://github.com/autonomousvision/carla_garage/tree/leaderboard_2

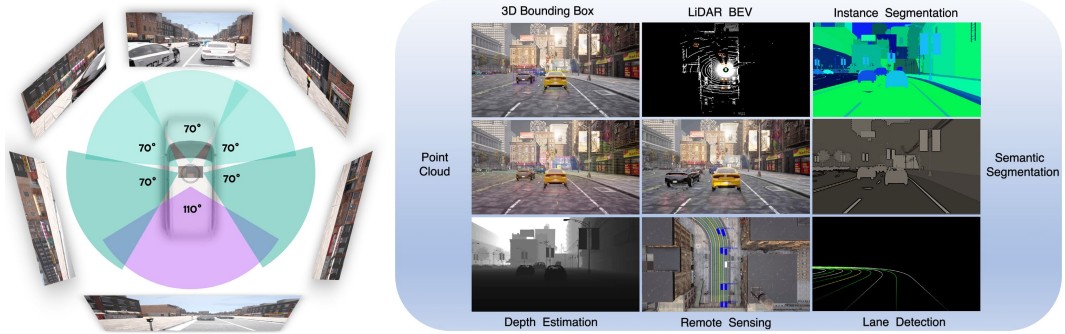

Figure 3: **Sensor setting and annotations of the expert dataset.** We follow the sensor settings of nuScenes [21]. The annotations include 3D bounding boxes, depth, semantic/instance segmentation, HD-Map, and RL value estimations and features from Think2Drive [31] Expert.

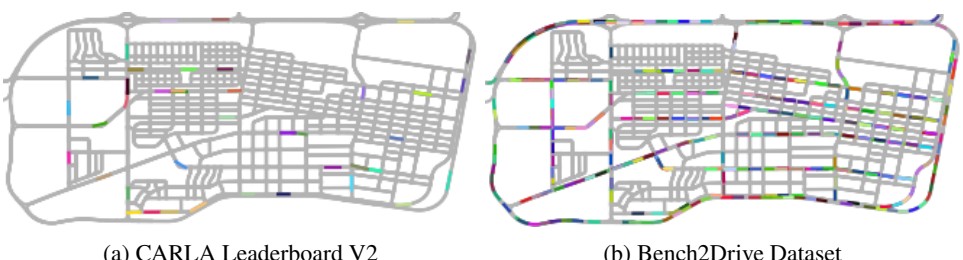

(a) CARLA Leaderboard V2         (b) Bench2Drive Dataset

Figure 4: **Distribution of scenario 'ConstructionObstacle' in Town12** We use different colors to represent different routes containing 'ConstructionObstacle'. Bench2Drive has more locations that be able to generate 'ConstructionObstacle'.

of these datasets significantly influence performance, rendering fair algorithm-level comparisons challenging. To address this, we have constructed a large-scale expert dataset with comprehensive annotations including 3D bounding boxes, depth, and semantic segmentation, sampled at 10 Hz, to serve as the official training set. As the information from expert could be an important guidance of student models [49, 27, 60], we also provide the expert model - Think2Drive's [31] value estimation and features. Fig. 3 gives an overview. To facilitate the re-implementation of existing E2E-AD methods of community, we adopt a sensor configuration similar to nuScenes [21]:

- 1x **LiDAR**: 64 channels, 85-meter range, 600,000 points per second
- 6x **Camera**: Surround coverage, 900x1600 resolution, JPEG compression (quality-level 20)
- 5x **Radar**: 100-meter range, 30° horizontal and vertical FoV
- 1x **IMU & GNSS**: Location, yaw, speed, acceleration, and angular velocity
- 1x **BEV Camera**: Debugging, visualization, remote sensing
- **HD-Map**: Lanes, centerlines, topology, dynamic light states, trigger areas for lights and stop signs

Moreover, to tackle the challenge posed by the long-tail distribution of data from both perception and behavior perspectives, a significant bottleneck in autonomous driving [62] (approximately 75% of the clips in nuScenes only involve the ego vehicle driving straight), we ensure the distribution of weather conditions, landscapes, and behaviors are as uniform as possible. We add more available locations for scenarios compared to the official routes of CARLA Leaderboard V2 as shown in Fig. 4, enhancing the data diversity. Further, we design 5 more scenarios beyond Leaderboard V2 to enhance behavior diversity as detailed in Appendix G. We give the distribution of scenarios and weathers and towns in Appendix B. As illustrated, Bench2Drive dataset is rich in both perception and behavior diversity.

For data partitioning, we segmented the driving process into short clips, each approximately 150 meters in length and containing a single specific scenario. This segmentation allows for the curriculum learning [63] of individual driving skills. To cater to different computational capabilities, we designed three data subsets: mini (10 clips for debugging and visualization), base (1,000 clips, comparable to nuScenes, suitable for 8xRTX3090 server), and full (10,000 clips for large-scale studies).

Table 2: **Skill Set & Scenarios**

| Skill | Scenario |
|---|---|
| Merging | CrossingBicycleFlow, EnterActorFlow, HighwayExit, InterurbanActorFlow, HighwayCutIn, InterurbanAdvancedActorFlow, MergerIntoSlowTrafficV2, MergeIntoSlowTraffic, NonSignalizedJunctionLeftTurn, NonSignalizedJunctionRightTurn, NonSignalizedJunctionLeftTurnEnterFlow, ParkingExit, LaneChange, SignalizedJunctionLeftTurn, SignalizedJunctionRightTurn, SignalizedJunctionLeftTurnEnterFlow |
| Overtaking | Accident, AccidentTwoWays, ConstructionObstacle, ConstructionObstacleTwoWays, HazardAtSideLaneTwoWays, HazardAtSideLane, ParkedObstacleTwoWays, ParkedObstacle, VehicleOpenDoorTwoWays |
| Emergency Brake | BlockedIntersection, DynamicObjectCrossing, HardBreakRoute, OppositeVehicleTakingPriority, OppositeVehicleRunningRedLight, ParkingCutIn, PedestrianCrossing, ParkingCrossingPedestrian, StaticCutIn, VehicleTurningRoute, VehicleTurningRoutePedestrian, ControlLoss |
| Give Way | InvadingTurn, YieldToEmergencyVehicle |
| Traffic Sign | EnterActorFlow, CrossingBicycleFlow, NonSignalizedJunctionLeftTurn, NonSignalizedJunctionRightTurn, NonSignalizedJunctionLeftTurnEnterFlow, OppositeVehicleTakingPriority, OppositeVehicleRunningRedLight, PedestrianCrossing, SignalizedJunctionLeftTurn, SignalizedJunctionRightTurn, SignalizedJunctionLeftTurnEnterFlow, TJunction, VanillaNonSignalizedTurn, VanillaSignalizedTurnEncounterGreenLight, VanillaSignalizedTurnEncounterRedLight, VanillaNonSignalizedTurnEncounterStopsign, VehicleTurningRoute, VehicleTurningRoutePedestrian |

## 3.3 Multi-Ability Evaluation

Existing planning benchmarks [28, 38, 37] assess the performance of AD systems by averaging scores across all provided routes. This approach offers a general overview of driving capabilities but fails to pinpoint specific strengths and weaknesses of different methods. Even worse, existing benchmarks in CARLA like Longest6 [6] and Leaderboard V2 [28] cover several kilometers, leading to high variance in the driving score metric. This variance arises because the infraction score penalizes errors through cumulative multiplication, which can significantly skew results. For instance, consider three test runs where each achieves 90% route completion, but the number of red lights run differs: 0, 1, and 2. The corresponding driving scores would be 90, $90 * 0.7 = 63$, and $90 * 0.7 * 0.7 = 44.1$, which causes a large standard deviation - 18.9 and thus makes comparison between methods unreliable.

To address these issues, we propose a more granular evaluation framework for all 44 scenarios by designing 5 distinct short routes (around 150 meters in length) per scenario, each featuring different weathers and towns, which result in a total of 220 routes. This approach allows people to assess AD systems' capabilities by isolated skills, leading to a more detailed analysis with reduced variance. Further, we summarize 5 advanced skills for urban driving: Merging, Overtaking, Give Way, Traffic Sign, Emergency Brake as in Table 2 and report the score of each skill. The decoupled design provides a clearer insight into which skills are effectively handled by the AD systems and which are not, fostering a more nuanced understanding of system performance.

Formally, the evaluation set consists of 220 routes and each route defines a pair of source location $(x_{src}, y_{src})$ and destination location $(x_{dst}, y_{dst})$ in one specific town and weather. Given raw sensor inputs (cameras, LiDAR, IMU/GPS, etc) as well as the target waypoints, the ego vehicle should drive from source to the destination location. We design two metrics to evaluate the performance:

- **Success Rate (SR)**: This metric measures the proportion of successfully completed routes within the allotted time and without traffic violations. A route is deemed successful if the ego vehicle reaches its destination without any rule infractions. The success rate is calculated as the ratio of successful routes to the total number of routes, as shown in Equ. 1 (left).
- **Driving Score (DS)**: This metric follows CARLA [28] official metric as reference. It considers both route completion and penalty for infractions. Specifically, it averages the route completion

percentages and penalizes infractions based on their severity, as depicted in Equation 1 (right). The driving score is normalized by the total number of routes from same type or group as well.

$$\text{Success Rate} = \frac{n_{\text{success}}}{n_{\text{total}}} \qquad \text{Driving Score} = \frac{1}{n_{\text{total}}} \sum_{i=1}^{n_{\text{total}}} \text{Route-Completion}_i * \prod_{j=1}^{n_{i,\text{penalty}}} p_{i,j} \quad (1)$$

where $n_{\text{success}}$ and $n_{\text{total}}$ denote the number of successful routes and total samples respectively; Route-Completion$_i$ representats the percentage of route distance completed for the $i$-th route; $p_{i,j}$ means the $j$-th infraction penalty on the $i$-th route. Please refer to Appendix F for details about infraction types and penalties scores.

Further, beyond the the goal achieving ability of algorithms, we propose the following two metrics to measure the efficiency and smoothness of driving trajectories:

- **Efficiency**: The CARLA team has implemented a function to check whether the self-driving car's speed is too low. This is determined by comparing the vehicle's speed with nearby vehicle:

$$\text{Speed Percentage} = \frac{\text{Ego Vehicle's Speed}}{\text{Average Speed of Nearby Vehicles}} \qquad (2)$$

This function calculates the speed percentage using the vehicle's speed and the average speed of nearby vehicles at current frame. CARLA Leaderboard sets four checkpoints per route and checks the ego vehicle's speed when the ego vehicle arrives a checkpoint. Specifically, if the vehicle is faster than nearby vehicles, the driving efficiency would be larger than 100%. The check results are included as a penalty in the final driving score. However, with only four checkpoints, the vehicle must cover 25% of the total route distance before reaching the next checkpoint. This leads to a high variance in the penalty values for low speeds, complicating the reflection of driving capabilities in the driving scores. To alleviate this, we increase the number of checkpoints to 20. Speed check is now performed every 5% of the total route length, and it is excluded from the driving score calculation. The final driving efficiency metric is defined as the average of the speed percentage over all checks.

$$\text{Driving Efficiency} = \frac{\sum_i \text{Speed Percentage}_i}{\text{Speed Check Times}} \qquad (3)$$

If the ego vehicle fails to pass the initial 5% checkpoint, this route is not included in the final driving efficiency metric calculation. To account for cases where abnormal speed spikes may occur (e.g., when the vehicle falls off the current map layer), speed percentage values exceeding 1000% are filtered out.

**Comfortness**: Comfortness is closely related to human experience and thus requires comparing autonomous driving policy with the behavior of numerous human driving experts to measure it. For this, we follow the popular benchmark nuPlan's [38] smoothness(also called comfort) protocol, which evaluates ego's minimum and maximum longitudinal accelerations, the maximum absolute values of lateral acceleration, yaw rate, yaw acceleration, the longitudinal component of jerk, and the maximum magnitude of the jerk vector. These variables are compared to thresholds with default values determined empirically from the examination of nuPlan's human expert trajectories. Comfortness is measured based on whether these values fall within the upper and lower bounds of the expert values.

$$\text{Frame Variable Smoothness (FVS)} = \begin{cases} \text{True} & \text{if lower bound} \leq p_i \leq \text{upper bound,} \\ \text{False} & \text{otherwise} \end{cases} \qquad (4)$$

$$p \in \text{smoothness vars}, \ 0 \leq i \leq \text{total frames}$$

where smoothness variables(vars) include: longitudinal acceleration - expert bound: [-4.05, 2.40], maximum absolute lateral acceleration - expert bound: [-4.89, 4.89], yaw rate - expert bound: [-0.95, 0.95], yaw acceleration - expert bound: [-1.93, 1.93], longitudinal component of jerk - expert bound: [-4.13, 4.13], maximum magnitude of jerk vector - expert bound:[-8.37, 8.37].
A trajectory is deemed Smooth only if all smoothness variables meet the smoothness criteria.

$$\text{Trajectory Smoothness} = \bigwedge_{i=0}^{\text{total frames}} \text{FVS}$$

In nuPlan, smoothness is determined by frame-by-frame evaluation of these variables over the entire trajectory, which makes it susceptible to local driving behaviors. For example, if a vehicle ahead suddenly brakes, the ego vehicle must also brake abruptly to avoid a collision. Even if the ego's hard brake behavior is appropriate in this case and its driving is smooth at other times, the entire trajectory could still be judged as unsmooth, leading to unreasonable evaluation results. To mitigate this issue, we segment the entire trajectory at a timestep interval $n = 20$ for evaluation.

$$\text{Segment Smoothness} = \bigwedge_{i=\text{start frame}}^{\text{end frame}} \text{FVS}$$

The final smoothness metric is defined as the ratio of smooth trajectory segments to the total number of segments.

$$\text{Smoothness} = \frac{\text{Number of Smoothness Segments}}{\text{Total Segments}}$$

Specifically, if the ego vehicle is blocked (speed remains below 0.1 for more than 60 seconds.), resulting in a failure case, this segment is still be considered as smooth because its speed is safe for human. Note that if the total frames of a trajectory are less than 20, the respective route is excluded from the smoothness assessment.

## 4 Experiments

### 4.1 Baselines & Datasets

To establish a starting point for the community, we have implemented several classic E2E-AD methods in Bench2Drive including:

- **UniAD** [4] explicitly conducts perception and prediction and uses Transformer Query to transport information. Together with it, we also implement the commonly used BEVFormer [10] in Bench2Drive,
- **VAD** [22] also adopts Transformer Query yet with vectorized scene representation and thus improves efficiency.
- **AD-MLP** [23] simply feeds the ego vehicle's history states into an MLP to predict future trajectories, which is a simple baseline for history state interpolation planner.
- **TCP** [64] only uses the front cameras and the ego state as inputs to predict both trajectories and control signals. It is a simple yet effective baseline in CARLA v1.
- **ThinkTwice** [30] promotes the idea of coarse-to-fine by refining the planning routes in a layer-by-layer manner and distilling the expert features.
- **DriveAdapter** [27] proposes a new paradigm to fully unleash the power of expert model by decoupling the learning of perception and planning and connecting the two parts by adapter modules.

Recognizing the varied computational resources available within the community, we have trained these baseline models on the *base* subset (1,000 clips). We use 950 clips for training while leaving 50 clips for open-loop evaluation. We ensure that the validation set contains at least one clip for each of 44 scenarios and the weather distribution is balanced. AD-MLP and TCP are trained with 1 * A6000 while ThinkTwice, DriveAdapter, UniAD, and VAD are trained with 8 * A100. For the closed-loop evaluation, we run all models in CARLA with the 220 test routes mentioned in Sec. 3.3 and calculate the metric accordingly. Note that some models' might have wrong behaviors in some certain routes (e.g., driving to some buggy location) and cause CARLA to crash without scoring. We treat these routes as 0 score. Please refer to Appendix C for more implementation details.

### 4.2 Results

In Table 3 and Table 4, we compare baselines E2E-AD methods with both open-loop and closed-loop evaluation, which lead to the following findings:

**Open-loop metric could indicate model convergence but it fails for advanced comparison.**. AD-MLP has a high L2 error and performs extremely bad in closed-loop evaluation while VAD has a low L2 error and a decent closed-loop performance. *It shows that we could use L2 error to verify the*

Table 3: **Open-loop and Closed-loop Results of E2E-AD Methods in Bench2Drive** under **base** training set. Avg. L2 is averaged over the predictions in 2 seconds under 2Hz, similar to UniAD. * denotes expert feature distillation.

| Method | Open-loop Metric | Closed-loop Metric | | | |
|---|---|---|---|---|---|
| | Avg. L2 ↓ | Driving Score ↑ | Success Rate(%) ↑ | Efficiency ↑ | Comfortness ↑ |
| AD-MLP [23] | 3.64 | 18.05 | 0.00 | 48.45 | 22.63 |
| UniAD-Tiny [4] | 0.80 | 40.73 | 13.18 | 123.92 | 47.04 |
| UniAD-Base [4] | **0.73** | **45.81** | **16.36** | 129.21 | 43.58 |
| VAD [22] | 0.91 | 42.35 | 15.00 | **157.94** | **46.01** |
| TCP* [5] | 1.70 | 40.70 | 15.00 | 54.26 | 47.80 |
| TCP-ctrl* | - | 30.47 | 7.27 | 55.97 | **51.51** |
| TCP-traj* | 1.70 | 59.90 | 30.00 | 76.54 | 18.08 |
| TCP-traj w/o distillation | 1.96 | 49.30 | 20.45 | **78.78** | 22.96 |
| ThinkTwice* [30] | **0.95** | 62.44 | 31.23 | 69.33 | 16.22 |
| DriveAdapter* [27] | 1.01 | **64.22** | **33.08** | 70.22 | 16.01 |

Table 4: **Multi-Ability Results of E2E-AD Methods** under **base** training set. * denotes expert feature distillation.

| Method | Ability (%) ↑ | | | | | |
|---|---|---|---|---|---|---|
| | Merging | Overtaking | Emergency Brake | Give Way | Traffic Sign | **Mean** |
| AD-MLP [23] | 0.00 | 0.00 | 0.00 | 0.00 | 4.35 | 0.87 |
| UniAD-Tiny [4] | 8.89 | 9.33 | 20.00 | 20.00 | 15.43 | 14.73 |
| UniAD-Base [4] | 14.10 | 17.78 | 21.67 | 10.00 | 14.21 | 15.55 |
| VAD [22] | 8.11 | 24.44 | 18.64 | 20.00 | 19.15 | **18.07** |
| TCP* [5] | 16.18 | 20.00 | 20.00 | 10.00 | 6.99 | 14.63 |
| TCP-ctrl* | 10.29 | 4.44 | 10.00 | 10.00 | 6.45 | 8.23 |
| TCP-traj* | 8.89 | 24.29 | **51.67** | 40.00 | 46.28 | 34.22 |
| TCP-traj w/o distillation | 17.14 | 6.67 | 40.00 | **50.00** | 28.72 | 28.51 |
| ThinkTwice* [30] | 27.38 | 18.42 | 35.82 | **50.00** | 54.23 | 37.17 |
| DriveAdapter* [27] | **28.82** | **26.38** | 48.76 | **50.00** | 56.43 | **42.08** |

*convergence and fitting status of neural networks*, i.e., when the L2 error is very high, there should be something wrong within the system. In this case, AD-MLP does not use raw sensors, which is similar to drive blindly and thus infeasible to fit the dataset. Notably, different from findings in nuScenes [21], AD-MLP fails to achieve decent L2 error in Bench2Drive, due to the better behavior diversity as shown in Fig. 5. On the other hand, UniAD-base has a lower L2 error compared to VAD yet with worse closed-loop performance, aligning with findings in [19, 24]. Open-loop evaluation ignore the issues including distribution shift [25] and causal confusion [26, 27] and thus fails to give meaningful comparsion for models with good fitting of dataset, demonstrating the importance of closed-loop evaluation. For efficiency and smoothness, we could observe that AD-MLP has the lowest efficiency due to its quick failure and stuck. UniAD has higher efficiency and smoother trajectories compared to TCP-traj, demonstrating the effectivenes of the UniAD's post optimization for the planning head.

**Expert feature distillation offers important guidance.** As pointed out in [49, 50], due to the high-dimensional input space of AD, i.e., multiple images and point clouds, E2E-AD methods tend to overfit. The features from expert, which already possesses strong driving knowledge, could be helpful to mitigate the issue by distillation. As a result, methods (TCP/ThinkTwice/DriveAdapter) with expert feature distillation outperforms those without (VAD/UniAD) by a large margin. From the comparison between TCP-traj with and without distillation, we could observe similar trend. However, in the real world setting, it could be difficult to obtain expert features, which worths further study.

**Interactive behaviors are difficult to learn.** All models' scores of skills regarding strong interaction (Merging, Overtaking, and Emergency Brake) are unsatisfying. It might come from two perspectives: (I) Long-tail issue. Even though we ensure that the number of clips for different scenarios are similar, there are only a few frames within one clip are about interactive behaviors. As a result, it might be challenging for the learning. (II) Imitation learning paradigm. Direct supervised training of control signals or trajectories might fail to give guidance regarding the gaming, thinking, and reasoning process of interaction. More advanced training paradigms could be a promising direction.

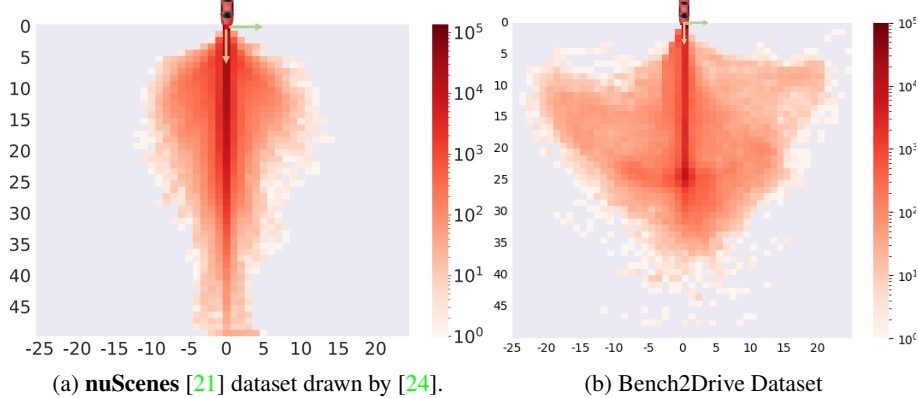

(a) **nuScenes** [21] dataset drawn by [24].          (b) Bench2Drive Dataset

Figure 5: **Distribution of ego vehicle's future location.** Bench2Drive possesses more turning trajectories, indicating better action diversity and thus providing better training data and having less gap between open-loop and closed-loop evaluation.

### 4.3 Case Analysis

We conduct visualizations and upload the results to https://github.com/Thinklab-SJTU/Bench2DriveZoo/blob/uniad/vad/analysis/analysis.md. For all five abilities, we choose some representative scenarios to visualize, where some baselines success and some baselines fail for the ease of comparison and analysis. We give the corresponding failure analysis so that the users and practioners could have a sense about the pros, cons, and future works of existing E2E-AD methods.

## 5   Conclusion

In this work, we present Bench2Drive, a new benchmark tailed for closed-loop evaluation of end-to-end autonomous driving methods. We open source a fully-annotated large-scale dataset as the official training set and a multi-ability evaluation toolkit for the granular driving skill assessment. State-of-the-art E2E-AD methods are tested in Bench2Drive with their pros and cons evaluated, which provides insights for the future direction.

**Limitations**: Since the rendering of simulation in CARLA has gaps compared to real world, utilizing real world datasets could be complementary as done in the concurrent work - NAVSIM [65]. Actually, there is a dilemma in the field for the evaluation of end-to-end autonomous driving algorithms:

| Source of Images | Pros | Cons |
|---|---|---|
| Real World Datasets | Realistic | Non-Reactive |
| Simulation Rendering | Reactive | Cartoon Style |

Generative models like diffusion models [66] might have the potential to provide realistic and reactive rendering, with some pioneering works in the field [67–69]. However, the illusion and artifact issue of diffusion requires further exploration.

**Social Impact**: The deployment of AD systems holds immense potential to revolutionize transportation, but it also brings significant ethical and safety concerns. Bench2Drive could serve as a platform for rigorously validating the capabilities of AD systems in a controlled and simulated environment, helping to identify potential flaws before real-world deployment. One of the primary risks is the simulation-reality gap—the difference between how an AD system performs in simulation versus in the real world. Simulations have the difficulties to fully replicate the complexities and unpredictability of real-world driving conditions. There is a risk that an AD system might perform well in simulation but fail in real-world scenarios due to unmodeled factors like rare edge cases, unexpected human behaviors, or varying environmental conditions. Bench2Drive is intended to complement, not replace, real-world testing, and it is crucial to emphasize that simulation is one part of a broader validation process that must include extensive on-road testing.

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

## A  Details of Data Collecting

The collection of data is a mix of automatic pipelines and manual checking. We give details below:

**Route**: We use the expert model Think2Drive to run on the predefined route files and only keep those without infractions. We design a traversal algorithm over all maps to determine whether a scenario could be triggered, aiming to cover towns as much as possible. The balance of weather, towns, and scenarios are ensured by manually checking. The behaviors and rendering of CARLA sometimes could be buggy as shown in Fig. 7 and we manually filter those bad clips.

**Annotations**: We utilize CARLA's official APIs to collect annotations. Notably, there are several bugs within the APIs: (I) All pedestrians' speed value are 0 from the API. We manually calculate their speed by differentiating during training of baseline methods. (II) The returned value of Speedometer and IMU could be None. We pad these values with 0 during training. (III) Some stop signs in CARLA are on the ground and thus there is no bounding box. To compensate this, We record all stop signs with rectangles to denote their trigger volume. (IV) Some static vehicles' rotation and location are wrong by API. Thus, we use the correct center and extent to obtain their 3D bounding boxes.

**Object Class**: Considering different attributes of different objects, we categorize all objects into four main types and store them group by group: Vehicle, Traffic Sign, Traffic Light, and Pedestrian. Vehicles are further subdivided into static and dynamic vehicles. Static vehicles remain stationary throughout the entire scenario and are distinguished by a unique actor identifier obtained from "static.prop.mesh". For Traffic Signs, they consist of speed_limit_sign, stop_sign, yield_sign, warning_sign (including warning construction, traffic warning, and warning accident), dirt_debris, and cone. Notably, signs involving trigger_volume, such as speed_limit_sign, stop_sign, and yield_sign, has trigger volume where we store the rectangle as well. The coordinates for warning signs and cones are obtained from the center and extent of their actor class, while dirt_debris requires additional conversion due to inaccurate coordinates. For Traffic Lights, the trigger volume coordinates for traffic signs and lights are relative to the actor, which requires extra transformation. Each traffic sign/light consists of two parts: the pole and the light/sign itself while the bounding boxes from API is only the lights/signs.

**Coordinate System**: Unlike the Y-down right-hand system used by Nuscenes, CARLA employs the Unreal Engine coordinate system, which is a Z-up left-hand coordinate system. In Compass, orientation with regard to the North ([0.0, -1.0, 0.0] in Unreal Engine) means the standard yaw angle in the left-hand system is theta in the compass minus 1/2 pi. In rare cases, this may result in NaN values, which need to be manually filtered.

**Map Information**: The HD-Map is organized into road_ids with lane_index. Each lane includes the world coordinates and orientations of points, lane type, color identifiers and adjacent road_ids-lane_ids (left, right and connected road ids and lane ids), as well as topological structures (e.g.,

Table 5: **Resource requirements for training on base set and evaluation on 220 routes.** The training of UniAD consists of BEVFormer, stage1, and stage2.

|  | Training | Evaluation |
|---|---|---|
| AD-MLP | 1 A600 * 1 day | 4 A6000 * 6 hours |
| TCP | 1 A600 * 1 day | 4 A6000 * 6 hours |
| VAD | 8 H800 * 5 days | 8 H800 * 2 days |
| UniAD-base | 8 H800 * (3+3+3)=9 days | 8 H800 * 2 days |

'Junction', 'Normal', 'EnterNormal', 'EnterJunction', 'PassNormal', 'PassJunction', 'StartJunction-MultiChange', or 'StartNormalMultiChange'). The Trigger_Volumes represent the trigger areas for signs, where 'Points' specify the vertices' locations of the trigger volume, 'Type' can be 'StopSign' or 'TrafficLight', and 'ParentActor_Location' provides details on the location of the parent actor associated with the trigger volume.

**Data Compression** To reduce file size, following [43], we adopted compressed data format. Images are compressed using JPG with the quality= 20. To avoid train-val gap during closed-loop evaluation, we also use in-memory JPG compression and decompression during inference. Semantic segmentation and depth data are stored as PNG files. We use a specialized algorithm called laszip to compress our LiDAR point clouds. JSON files are compressed using GZIP.

# B   Distribution of Scenarios, Towns and Weathers

As shown in Fig. 8, the distribution of weathers are nearly uniform while the distribution of town is dominated by Town12 and Town13. The imbalance comes from two perspectives: (I) New towns, like Town12 and Town13, are designed on purpose by CARLA team to be much larger than old towns so that the community could explore applications in city-level scene. Thus, we collect more data in larger towns for more diverse landscapes. (II) Lots of new scenarios in Leaderboard v2 are designed recently and thus old towns do not support the layouts required by new scenarios. For example, parting exit requires the existence of other parked vehicles. As a result, we have to collect more data in new towns to ensure the balance of scenario types.

# C   Implementation Details of Baselines

For all baseline E2E-AD methods, we strictly follow their official open-sourced code, environments, and configs. There are a few modifications: (I) For methods with object detection module, we change the detection classes according to CARLA[3]. (II) Data collection of Bench2Drive is in 10Hz while nuScenes with bounding boxes is in 2Hz. Due to Bench2Drive's longer clips, Bench2Drive-base has approximately 10x frames compared to nuScenes yet with higher redundancy. For computationally demanding methods like UniAD/VAD/ThinkTwice/DriveAdapter, we train $\frac{1}{10}$ epochs compared to the original version. We observe similar level of loss due to similar number of training steps. (III) Since ThinkTwice and DriveAdapter require expert's BEV features, we use Think2Drive expert to regenerate those expert feature. Additionally, for fair comparison with other methods, we modify both of them into 6 cameras without LiDAR.

# D   Training and Evaluation Resource Requirements

We report resource requirements of training and evaluating baselines. Note that the evaluation time could be linearly speed up with more GPUs to parally evaluate on more routes.

---

[3]https://carla.readthedocs.io/en/latest/catalogue_vehicles/

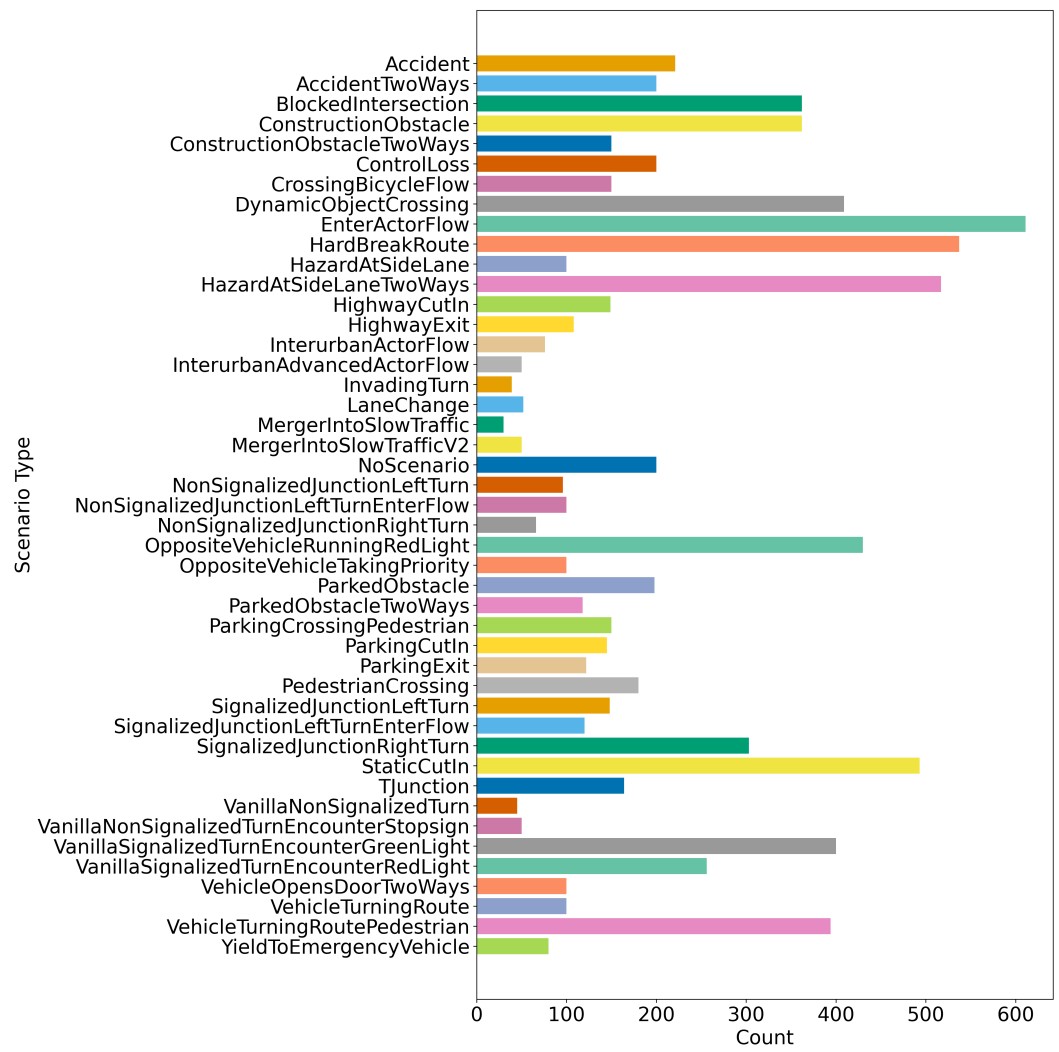

Figure 6: **Scenario Distribution of Bench2Drive Dataset**

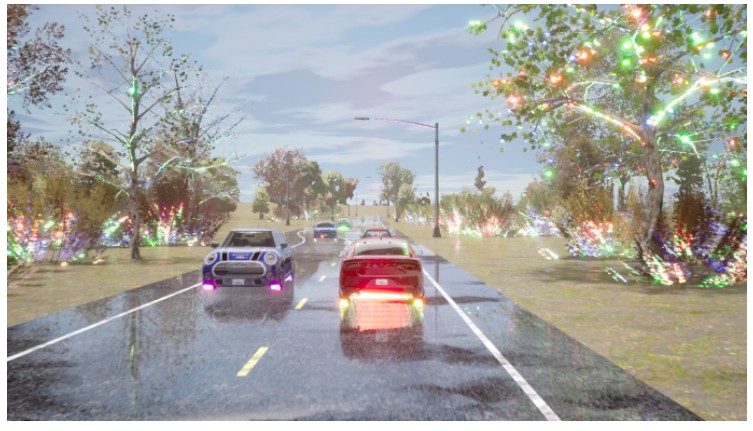

Figure 7: **Bugged Rendering of CARLA**.

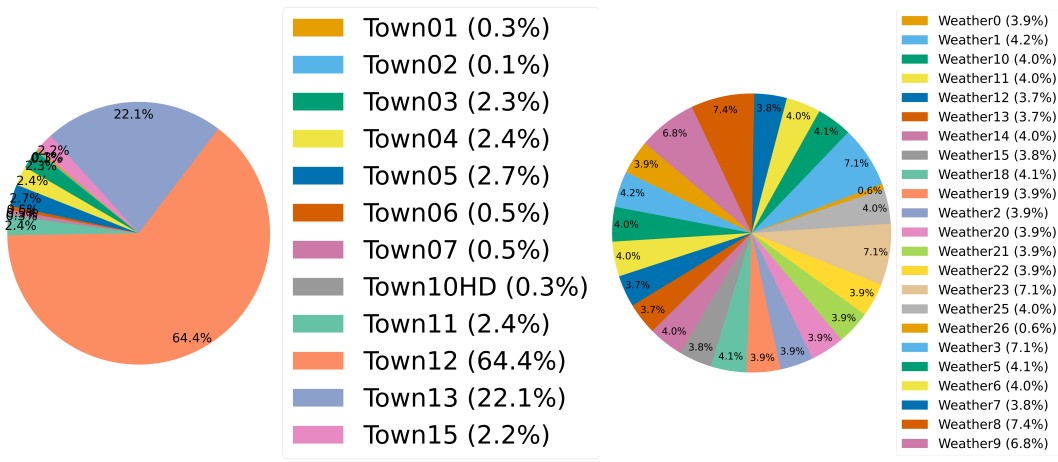

Figure 8: **Town and Weather Distribution of Bench2Drive Dataset**

# E Behavior Model of NPC Agents

In CARLA, three behavior types are preset in CARLA Agent: **cautious**, **normal** and **aggressive**. These behavior types govern the driving actions of NPCs, influencing factors such as speed, responses to other vehicles, and safety protocols. The key parameters for each behavior mode include:

- **max_speed**: Sets the maximum speed(km/h) that an NPC vehicle can reach.
- **speed_lim_dist**: Value in km/h that defines how far your vehicle's target speed will be from the current speed limit
- **speed_decrease**: Controls the deceleration of the NPC when approaching a slower vehicle ahead.
- **safety_time**: Estimates the time to a collision if the vehicle in front suddenly brakes.
- **min_proximity_threshold**: Defines the minimum distance before the NPC takes actions such as evasive maneuvers or tailgating.
- **braking_distance**: The distance at which the NPC performs an emergency stop to avoid a collision.
- **tailgate_counter**: A counter that prevents the NPC from initiating a new tailgating action too soon after the last one.

These behavior designs interact with the ego (self-driving) vehicle, ensuring that NPCs respond to the presence and actions of the ego vehicle in the simulation. The parameters for different behavior styles are as follows,

Table 6: Comparison of Cautious, Normal, and Aggressive behavior parameters.

| Parameter | Cautious | Normal | Aggressive |
|---|---|---|---|
| **max_speed (km/h)** | 40 | 50 | 70 |
| **speed_lim_dist (km/h)** | 6 | 3 | 1 |
| **speed_decrease (km/h)** | 12 | 10 | 8 |
| **safety_time (seconds)** | 3 | 3 | 3 |
| **min_proximity_threshold (meters)** | 12 | 10 | 8 |
| **braking_distance (meters)** | 6 | 5 | 4 |
| **tailgate_counter (times)** | 0 | 0 | -1 |

The rule-based behavior decision algorithm in behavior_agent.py for non-player character (NPC) vehicles is shown below:

The behavior of walkers/pedestrians is characterized by their consistent adherence to the route line at a constant speed, with the inability to walk backward. In autonomous driving scenarios, pedestrian

**Algorithm 1** Non-Player Character (NPC) Vehicles Behavior Decision

```
 1: Update vehicle's surrounding information
 2: if red light or stop sign detected then
 3:     return emergency_stop()
 4: end if
 5: if pedestrian detected and within braking distance then
 6:     return emergency_stop()
 7: end if
 8: if other vehicle nearby then
 9:     if within braking distance then
10:         return emergency_stop()
11:     else
12:         return car_following()
13:     end if
14: else if ego vehicle at intersection and (turn left or turn right) then
15:     adjust_speed_limit()
16:     return PID_Control
17: else if in normal driving conditions then
18:     maintain_speed_limit()
19:     return PID_Control
20: end if
```

Table 7: **Infraction Types & Penalty**. Following https://leaderboard.carla.org/.

| Infraction | Penalty | Note |
|---|---|---|
| Pedestrian Collision | 0.50 | Punished every infraction. |
| Vehicles Collision | 0.60 | Punished every infraction. |
| Other Collision | 0.65 | Punished every infraction. |
| Running Red Light | 0.70 | Punished every infraction. |
| Scenario Timeout | 0.70 | Fail to pass certain scenarios in 4 minutes. |
| Too Slow | 0.70 | Fail to maintain a suitable speed with surrounding vehicle. |
| No Give Way | 0.70 | Failure to yield to emergency vehicle. |
| Off-road | - | Not considered in route completion. |
| Route Deviation | - | Deviates more than 30 meters. Shutdown immediately. |
| Agent Blocked | - | No action for 180 seconds. Shutdown immediately. |
| Route Timeout | - | Exceed the maximum time limit. Shutdown immediately. |

safety is of utmost importance. Pedestrians have the highest priority when autonomous vehicles interact with humans, and autonomous vehicles should learn to give way to pedestrians regardless of traffic conditions.

## F  Details about Infraction Score

In Table 7, we gives the penalty score of each infraction designed by Leaderboard v2.

## G  Description of Scenarios

Bench2Drive provides 44 corner scenarios, extended from CARLA Leaderboard V2. We give details about them below:

*1. ControlLoss*

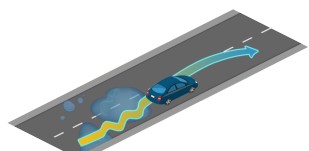

The ego vehicle loses control due to bad conditions on the road and it must recover, coming back to its original lane.

**2. ParkingExit**

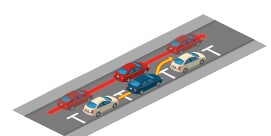

The ego vehicle must exit a parallel parking bay into a flow of traffic.

**3. ParkingCutIn**

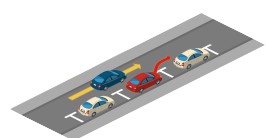

The ego vehicle must slow down or brake to allow a parked vehicle exiting a parallel parking bay to cut in front.

**4. StaticCutIn**

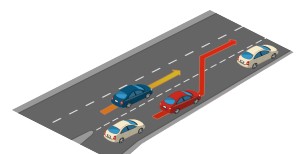

The ego vehicle must slow down or brake to allow a vehicle of the slow traffic flow in the adjacent lane to cut in front. Compared to *ParkingCutIn*, there are more cars in the adjacent lane and any one of them may cut in.

**5. ParkedObstacle**

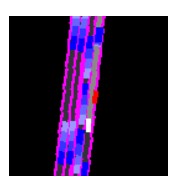

The ego vehicle encounters a parked vehicle blocking part of the lane and must perform a lane change into traffic moving in the same direction to avoid it.

**6. ParkedObstacleTwoWays**

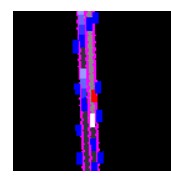

The '*TwoWays*' version of *ParkedObstacle*. The ego vehicle encounters a parked vehicle blocking the lane and must perform a lane change into traffic moving in the opposite direction to avoid it.

**7. Construction**

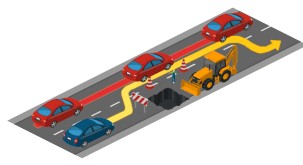

The ego vehicle encounters a construction site blocking and must perform a lane change into traffic moving in the same direction to avoid it. Compared to *ParkedObstacle*, the construction occupies more width of the lane. The ego vehicle has to completely deviate from its task route temporarily to bypass the construction zone.

**8. ConstructionTwoWays**

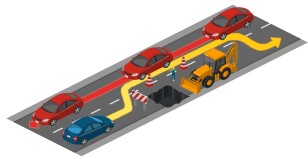

The '*TwoWays*' version of *Construction*.

**9. Accident**

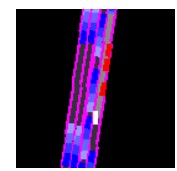

The ego vehicle encounters multiple accident cars blocking part of the lane and must perform a lane change into traffic moving in the same direction to avoid it. Compared to *ParkedObstacle* and *Construction*, these accident cars occupy more length along the lane. The ego vehicle has to completely deviate from its task route for a longer time to bypass the accident zone.

**10. AccidentTwoWays**

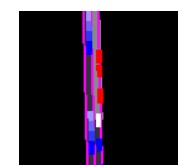

The '*TwoWays*' version of *Accident*. Compared to *ParkedObstacleTwoWays* and *ConstructionTwoWays*, there is a much shorter time window for the ego vehicle to bypass the route obstacles (i.g. accident cars).

**11. HazardAtSideLane**

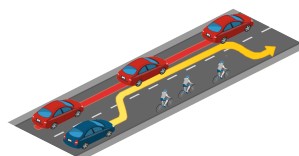

The ego vehicle encounters a slow-moving hazard blocking part of the lane. The ego vehicle must brake or maneuver next to a lane of traffic moving in the same direction to avoid it.

**12. HazardAtSideLaneTwoWays**

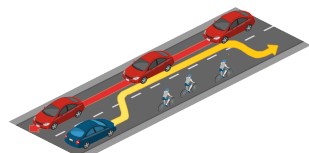

The ego vehicle encounters a slow-moving hazard blocking part of the lane. The ego vehicle must brake or maneuver to avoid it next to a lane of traffic moving in the opposite direction.

**13. VehiclesDooropenTwoWays**

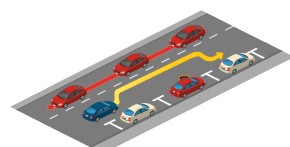

The ego vehicle encounters a parked vehicle opening a door into its lane and must maneuver to avoid it.

**14. DynamicObjectCrossing**

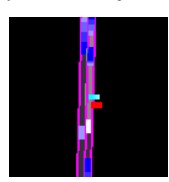

A walker or bicycle behind a static prop crosses the road suddenly when the ego vehicle is close to the prop. The ego vehicle must make a hard brake promptly.

**15. ParkingCrossingPedestrian**

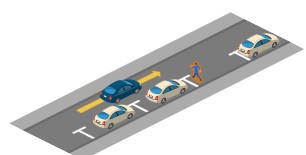

The ego vehicle encounters a pedestrian emerging from behind a parked vehicle and advancing into the lane. The ego vehicle must brake or maneuver to avoid it. Compared to *DynamicObjectCrossing*, the pedestrian is closer to the road and the ego vehicle has to act more timely.

**16. HardBrake**

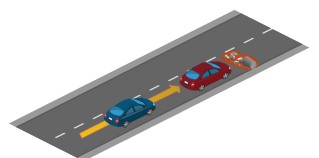

The leading vehicle decelerates suddenly and the ego vehicle must perform an emergency brake or an avoidance maneuver.

**17. YieldToEmergencyVehicle**

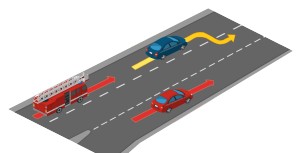

The ego vehicle is approached by an emergency vehicle coming from behind. The ego vehicle must maneuver to allow the emergency vehicle to pass.

## 18. *InvadingTurn*

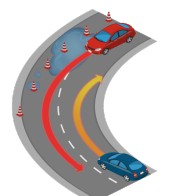

When the ego vehicle is about to turn right, a vehicle coming from the opposite lane invades the ego's lane, forcing the ego to move right to avoid a possible collision.

## 19. *PedestrainCrossing*

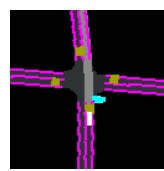

While the ego vehicle is entering a junction, a group of natural pedestrians suddenly cross the road and ignore the traffic light. The ego vehicle must stop and wait for all pedestrians to pass even though there is a green traffic light or a clear junction.

## 20. *VehicleTurningRoutePedestrian*

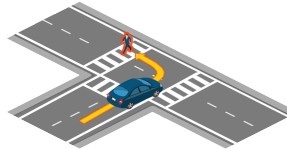

While performing a maneuver, the ego vehicle encounters a pedestrian crossing the road and must perform an emergency brake or an avoidance maneuver.

**21. *VehicleTurningRoute*** While performing a maneuver, the ego vehicle encounters a bicycle crossing the road and must perform an emergency brake or an avoidance maneuver. Compared to *VehicleTurningRoutePedestrian*, the bicycle moves faster and the ego has to brake earlier.

## 22. *BlockedIntersection*

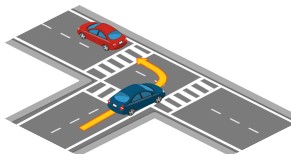

While performing a maneuver, the ego vehicle encounters a stopped vehicle on the road and must perform an emergency brake or an avoidance maneuver.

## 23. *SignalizedJunctionLeftTurn*

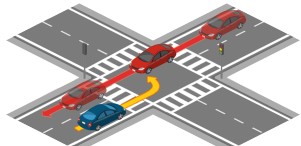

The ego vehicle is performing an unprotected left turn at an intersection, yielding to oncoming traffic.

## 24. *SignalizedJunctionLeftTurnEnterFlow*

The ego vehicle is performing an unprotected left turn at an intersection, merging into opposite traffic.

## 25. *NonSignalizedJunctionLeftTurn*

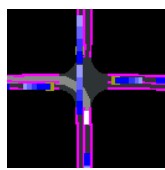

Non-signalized version of *SignalizedJunctionLeftTurn*. The ego has to negotiate with the opposite vehicles without traffic lights.

## 26. *NonSignalizedJunctionLeftTurnEnterFlow*

Non-signalized version of *SignalizedJunctionLeftTurnEnterFlow*.

### 27. *SignalizedJunctionRightTurn*

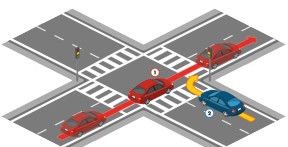

The ego vehicle is turning right at an intersection and has to safely merge into the traffic flow coming from its left.

### 28. *NonSignalizedJunctionRightTurn*

Non-signalized version of *SignalizedJunctionRightTurn*. The ego has to negotiate with the traffic flow without traffic lights.

### 29. *EnterActorFlows*

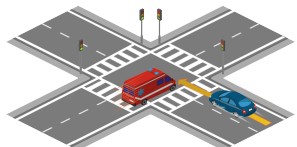

A flow of cars runs a red light in front of the ego when it enters the junction, forcing it to react (interrupting the flow or merging into the flow). These vehicles are 'special' ones such as police cars, ambulances, or firetrucks.

### 30. *HighwayExit*

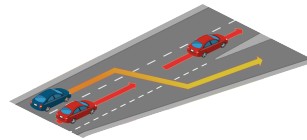

The ego vehicle must cross a lane of moving traffic to exit the highway at an off-ramp.

### 31. *MergerIntoSlowTraffic*

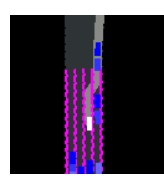

The ego vehicle must merge into a slow traffic flow on the off-ramp when exiting the highway.

### 32. *MergerIntoSlowTrafficV2*

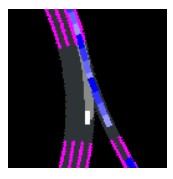

The ego vehicle must merge into a slow traffic flow coming from the on-ramp when driving on highway roads.

### 33. *InterurbanActorFlow*

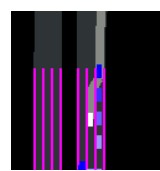

The ego vehicle leaves the interurban road by turning left, crossing a fast traffic flow.

### 34. *InterurbanAdvancedActorFlow*

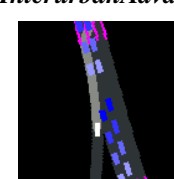

The ego vehicle incorporates into the interurban road by turning left, first crossing a fast traffic flow, and then merging into another one.

### 35. *HighwayCutIn*

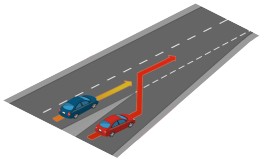

The ego vehicle encounters a vehicle merging into its lane from a highway on-ramp. The ego vehicle must decelerate, brake, or change lanes to avoid a collision.

### 36. CrossingBicycleFlow

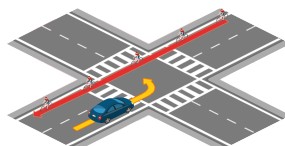

The ego vehicle needs to perform a turn at an intersection yielding to bicycles crossing from either the left.

### 37. OppositeVehicleRunningRedLight

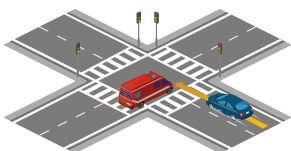

The ego vehicle is going straight at an intersection but a crossing vehicle runs a red light, forcing the ego vehicle to avoid the collision.

### 38. OppositeVehicleTakingPriority

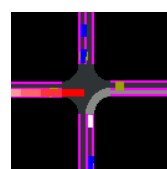

Non-signalized version of *OppositeVehicleTakingPriority*.

### 39. VinillaNonSignalizedTurn

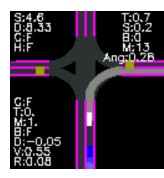

A basic scenario for the ego vehicle to learn to pass through a non-signalized junction (without traffic signs and traffic lights).

### 40. VinillaNonSignalizedTurnEncounterStopsign

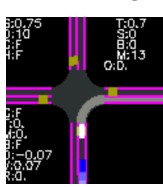

A basic scenario for the ego vehicle to learn to stop and start at stop signs.

### 41. VinillaSignalizedTurnEncounterGreenLight

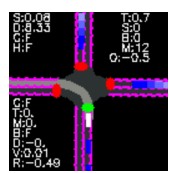

A basic scenario for the ego vehicle to learn to pass through the signalized junction.

### 42. VinillaSignalizedTurnEncounterRedLight

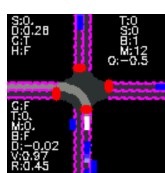

A basic scenario for the ego vehicle to learn to pass through the signalized junction when the traffic light changes from red to green.

### 43. LaneChange

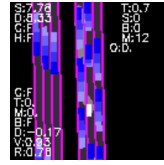

A basic scenario for the ego vehicle to learn to change lanes and avoid collision.

*44. TJunction*

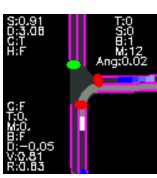

A basic scenario for the ego vehicle to learn to pass through a T-junction.

## H  Author Statement

We bear all responsibility in case of violation of rights, etc., and confirm the license of data, codes, and checkpoints as in Sec. 3.

## I  License

All data, codes, and checkpoints are in GitHub and Huggingface under Apache License 2.0.

## J  Datasheet

### J.1  Motivation

**For what purpose was the dataset created? Was there a specific task in mind? Was there a specific gap that needed to be filled? Please provide a description.** We build the benchmark to fulfill the need of comprehensive and realistic testing environments for Full Self-Driving (FSD). The primary task is end-to-end autonomous driving. Existing benchmarks failed to provide a closed-loop granular assessment of driving skills for E2E-AD methods.

**Who created the dataset (e.g., which team, research group) and on behalf of which entity (e.g., company, institution, organization)?** This dataset is curated by Xiaosong Jia, Zhenjie Yang, Qifeng Li, Zhiyuan Zhang, Junchi Yan from ReThinkLab with School of AI and Department of CSE, Shanghai Jiao Tong University.

### J.2  Distribution

**Will the dataset be distributed to third parties outside of the entity (e.g., company, institution, organization) on behalf of which the dataset was created?** Yes.

**How will the dataset be distributed (e.g., tarball on website, API, GitHub)?** All data, codes, and checkpoints are in GitHub (https://github.com/Thinklab-SJTU/Bench2Drive) and Huggingface (https://huggingface.co/datasets/rethinlab/Bench2Drive).

### J.3  Maintenance

**Who will be supporting/hosting/maintaining the dataset?** All authors and potentially new members of ReThinLab in Shanghai Jiao Tong University led by Prof. Junchi Yan.

**How can the owner/curator/manager of the dataset be contacted (e.g., email address)?** Please contact Xiasong Jia (jiaxiaosong@sjtu.edu.cn) and Junchi Yan (yanjunchi@sjtu.edu.cn).

**Is there an erratum?** No erratum as of submission.

**Will the dataset be updated (e.g., to correct labeling errors, add new instances, delete instances)?** Yes, we will maintain the dataset.

**Will older versions of the dataset continue to be supported/hosted/maintained?** Yes.

**If others want to extend/augment/build on/contribute to the dataset, is there a mechanism for them to do so?** Yes, they could follow the guide in `https://github.com/Thinklab-SJTU/Bench2Drive`.

## J.4 Composition

**What do the instances that comprise the dataset represent?** One basic instance is one clip. Each clip contains hundreds or thousands of frames in 10 Hz with raw sensor information and annotations..

**How many instances are there in total (of each type, if appropriate)?** There are 13638 clips in Bench2Drive.

**Are relationships between individual instances made explicit?** Yes. Each clip is annotated with scenario types and locations. Clips could have the same scenario type or nearby location.

**Are there recommended data splits (e.g., training, development/validation, testing)?** Yes. In the github repo.

**Is the dataset self-contained, or does it link to or otherwise rely on external resources?** Self-contained.

## J.5 Collection Process

**Who was involved in the data collection process (e.g., students, crowdworkers, contractors) and how were they compensated (e.g., how much were crowdworkers paid)?** All data is collected in CARLA automatically The collection code is written by authors.

## J.6 Use

**What (other) tasks could the dataset be used for?** With existing annotations, the dataset could also be used to conduct 3D object detection, semantic segmentation, instance segmentation, point cloud segmentation, depth estimation, tracking, motion prediction.

