# OpenReview forum: "Bench2Drive: Towards Multi-Ability Benchmarking of Closed-Loop End-To-End Autonomous Driving"
_NeurIPS.cc/2024/Datasets_and_Benchmarks_Track — NeurIPS 2024 Track Datasets and Benchmarks Poster_

### Official Review · Reviewer_K6WS · 2024-07-14
**Review Comments of Bench2Drive**

**Rating:** 6
**Confidence:** 4

**Review:**

Bench2Drive introduces a new benchmarking framework designed to evaluate the multiple abilities of end-to-end autonomous driving (E2E-AD) systems in a closed-loop manner.

It aims to overcome the limitations of current E2E-AD evaluation methods, which rely on open-loop metrics or fixed route testing and fail to capture comprehensive driving performance under diverse conditions. Bench2Drive provides a realistic, comprehensive, and fair testing environment by incorporating 44 interactive scenarios, varied weather conditions, and multiple locations. It evaluates specific driving skills across 220 short routes, allowing for a detailed analysis of different AD systems' capabilities.

The closed-loop evaluation protocol ensures that the AD system's actions directly influence the environment, offering an accurate assessment of driving performance. Several state-of-the-art E2E-AD methods are implemented and evaluated using Bench2Drive, providing insights into their strengths and weaknesses.

Despite its advantages, Bench2Drive is currently limited to the CARLA simulation environment, which may not fully capture real-world complexities. The reliance on simulated data might introduce biases, and the computational overhead of running extensive evaluations is not thoroughly discussed.

The manuscript could benefit from more detailed comparisons with existing benchmarks, a deeper analysis of computational complexity, and an expanded discussion of ethical considerations.

Overall, Bench2Drive represents a possible advancement in the evaluation of E2E-AD systems, providing a robust framework for benchmarking that addresses current gaps in the field. Addressing the comments in Opportunities for Improvement and Limitations could further improve the quality of this work.

**Strengths:**

(+) The introduction of Bench2Drive aims to address the gap in autonomous driving evaluation by providing a comprehensive and realistic closed-loop benchmarking framework, which is crucial for advancing the field of E2E-AD.

(+) The granular skill assessment allows for a detailed analysis of specific driving abilities, facilitating targeted improvements in AD algorithms. The implementation/benchmark of state-of-the-art E2E-AD methods is a plus of this work.

(+) The extensive coverage of diverse driving scenarios and conditions ensures a thorough evaluation of AD systems, highlighting their capabilities and limitations in various real-world situations.

**Additional Feedback:**

Overall, Bench2Drive represents a step forward in the evaluation of E2E-AD systems. The comprehensive scenario coverage, granular skill assessment, and closed-loop evaluation protocol provide a robust framework for benchmarking AD systems.

Addressing the identified opportunities for improvement and expanding the discussion of limitations and ethical considerations would further strengthen the manuscript and its contributions to the field.

**Clarity:**

The manuscript is somewhat clear and organized, with each section logically progressing to the next.

The descriptions of the benchmark's components, evaluation protocol, and experimental results are detailed and mostly easy to follow. Including more graphical illustrations and detailed captions for figures could further enhance clarity.

Additionally, the quality of existing figures should be further enhanced.

**Correctness:**

The manuscript presents a well-defined benchmarking framework, and the experimental results are supported by evaluations of several state-of-the-art E2E-AD methods. The claims are substantiated with detailed explanations and comparisons, ensuring, to a certain extent, the correctness of the findings.

**Documentation:**

The manuscript includes detailed documentation of the benchmark's design, data collection process, and evaluation protocol. The availability of the dataset, code, and evaluation toolkit on GitHub and Huggingface is commendable and facilitates reproducibility and community engagement.

**Ethics:**

The ethical considerations of deploying AD systems are briefly mentioned. A more thorough discussion of the potential societal impacts and safety considerations of using Bench2Drive for evaluating AD systems could be included to address ethical concerns comprehensively.

**Limitations:**

(-) The benchmark is currently limited to the CARLA simulation environment, which may not fully capture the complexities of real-world driving scenarios.

(-) The reliance on simulated data might introduce biases that are not present in real-world driving, necessitating careful interpretation of the benchmark results. Future work could explore integrating real-world data to enhance the realism of the evaluation.

(-) The potential computational overhead of running extensive closed-loop evaluations, especially for large-scale studies, is not thoroughly discussed.

(-) Other potential limitations as listed in Opportunities for Improvement.

**Opportunities For Improvement:**

- Additional experiments on real-world datasets or simulations that closely mimic real-world conditions could further validate the robustness and applicability of Bench2Drive.

- The discussion of potential limitations and challenges associated with the proposed benchmark, such as the gaps between simulation and real-world driving, should be expanded.

- The manuscript could benefit from a more in-depth analysis of the computational complexity and resource requirements for running the benchmark, particularly for large-scale studies.

- The technical novelty of the benchmark could be further emphasized by providing more detailed comparisons with existing benchmarks and highlighting the unique advantages of Bench2Drive. Since the overall dataset contains other annotations derived from CARLA, i.e., 3D bounding boxes, depth, semantic/instance segmentation, HD maps, and RL value estimations, it is highly recommended to include these tasks and establish benchmarks to further enhance the comprehensiveness of Bench2Drive.

- The quality of existing figures should be further enhanced. For example, the color sections in Fig. 2 and Fig. 4 are shown without specific definitions or explanations. The font size of the text in Fig. 3 is too small to match that of the main text. The two subfigures in Fig. 6 have different font sizes. Some figures, such as Fig. 5, take up quite a lot of space and are recommended to move to the appendix.

**Relation To Prior Work:**

The manuscript provides a rough review of existing benchmarks and positions Bench2Drive as a significant advancement over current methods. However, more explicit comparisons with closely related benchmarks and a detailed discussion of the unique contributions of Bench2Drive would strengthen the relation to prior work.

**Summary And Contributions:**

This manuscript introduces Bench2Drive, a new benchmarking framework designed to evaluate the multiple abilities of end-to-end autonomous driving (E2E-AD) systems in a closed-loop manner.

The primary motivation is to address the limitations of current E2E-AD evaluation methods, which typically rely on open-loop metrics or fixed route testing, failing to capture the comprehensive driving performance under diverse conditions and scenarios. Bench2Drive provides a more realistic, comprehensive, and fair testing environment by incorporating a wide range of interactive scenarios, weather conditions, and locations.

The authors claimed the contributions of this work as follows:
- Comprehensive Scenario Coverage: Bench2Drive includes 44 interactive driving scenarios under various weather conditions and locations, providing a thorough evaluation of E2E-AD systems.

- Granular Skill Assessment: The benchmark evaluates specific driving skills across 220 short routes, allowing for detailed analysis of different AD systems' capabilities.

- Closed-Loop Evaluation Protocol: Bench2Drive uses a closed-loop protocol where the AD system's actions directly influence the environment, offering an accurate assessment of driving performance.

- Implementation of Baseline Methods: Several state-of-the-art E2E-AD methods are implemented and evaluated using Bench2Drive, providing insights into their strengths and weaknesses.

---

> ### Author Rebuttal · Authors · 2024-08-19
>
> > **Q1: More results closely mimic real-world conditions**
>
> Following reviewer WLvp and your advice, we implement two additional metrics to evaluate two new perspectives of the driving to make the benchmark more practical and meaningful to the real world application:
>
> - **Efficiency**: The CARLA team has implemented a function to check whether the self-driving car's speed is too low. This is determined by comparing the vehicle's speed with nearby vehicle:
> $$\text{Speed Percentage}=\frac{\text{Ego Vehicle's Speed}}{\text{Average Speed of Nearby Vehicles}} $$
> This function calculates the speed percentage using the vehicle’s speed and the average speed of nearby vehicles at current frame. CARLA Leaderboard sets four checkpoints per route and checks the ego vehicle's speed when the ego vehicle arrives a checkpoint. Specifically, if the vehicle is faster than nearby vehicles, the driving efficiency would be larger than 100%. The check results are included as a penalty in the final driving score. However, with only four checkpoints, the vehicle must cover 25% of the total route distance before reaching the next checkpoint. This leads to a high variance in the penalty values for low speeds, complicating the reflection of driving capabilities in the driving scores. To alleviate this, we increase the number of checkpoints to 20. Speed check is now performed every 5% of the total route length, and it is excluded from the driving score calculation. The final driving efficiency metric is defined as the average of the speed percentage over all checks.
> $$\text{Driving Efficiency}=\frac{\sum_i \text{Speed Percentage}\phantom{}_i}{\text{Speed Check Times}} $$
> If the ego vehicle fails to pass the initial 5% checkpoint, this route is not included in the final driving efficiency metric calculation. To account for cases where abnormal speed spikes may occur (e.g., when the vehicle falls off the current map layer), speed percentage values exceeding 1000% are filtered out.
>
> - **Smoothness**: Smoothness is closely related to human experience and thus requires comparing autonomous driving policy with the behavior of numerous human driving experts to measure it. For this, we follow the popular benchmark nuPlan's smoothness(also called comfort) protocol, which evaluates ego’s minimum and maximum longitudinal accelerations, the maximum absolute values of lateral acceleration, yaw rate, yaw acceleration, the longitudinal component of jerk, and the maximum magnitude of the jerk vector. These variables are compared to thresholds with default values determined empirically from the examination of nuPlan’s human expert trajectories. Smoothness is measured based on whether these values fall within the upper and lower bounds of the expert values.
> $$\small \text{Frame Variable Smoothness(FVS)} = \begin{cases}
> \text{True} & \text{if lower bound}\leq p_{\,i} \leq \text{upper bound}, \\
> \text{False} & \text{otherwise}
> \end{cases} \\
>  p \in \text{smoothness vars}, \, 0\leq i \leq \text{total frames}$$
>
>     - smoothness variables(vars) include:
>       - `longitudinal_acceleration, expert bound:[-4.05, 2.40]`
>       - `maximum_absolute_lateral_acceleration, expert bound:[-4.89, 4.89]`
>       - `yaw_rate, expert bound:[-0.95, 0.95]`
>       - `yaw_acceleration, expert bound:[-1.93, 1.93]`
>       - `longitudinal_component_of_jerk, expert bound:[-4.13, 4.13]`
>       - `maximum_magnitude_of_jerk_vector, expert bound:[-8.37, 8.37]`
>
> A trajectory is deemed Smooth only if all smoothness variables meet the smoothness criteria.
> $$\text{Trajectory Smoothness} =\bigwedge_{i=0}^{\text{total frames}}\text{FVS}
> $$
> In nuPlan, smoothness is determined by frame-by-frame evaluation of these variables over the entire trajectory, which makes it susceptible to local driving behaviors. For example, if a vehicle ahead suddenly brakes, the ego vehicle must also brake abruptly to avoid a collision. Even if the ego’s hard brake behavior is appropriate in this case and its driving is smooth at other times, the entire trajectory could still be judged as unsmooth, leading to unreasonable evaluation results. To mitigate this issue, we segment the entire trajectory at a timestep interval $\small n=20$ for evaluation.
> $$
> \text{Segment Smoothness} =\bigwedge_{i=\text{start frame}}^{\text{end frame}}\text{FVS}
> $$
> The final smoothness metric is defined as the ratio of smooth trajectory segments to the total number of segments.
> $$
> \text{Smoothness}=\frac{\text{Number of Smoothness Segments}}{\text{Total Segments}}
> $$
> Specifically, if the ego vehicle is blocked (speed remains below 0.1 for more than 60 seconds.), resulting in a failure case, this segment is still be considered as smooth because its speed is safe for human. Note that if the total frames of a trajectory are less than 20, the respective route is excluded from the smoothness assessment.
>
>
> We are re-runing all baselines to get the two new metrics while some of them are not finished yet due to the heavy computational buderns and we will update the full results in the Github. The following are results of finished ones:
>
> | Method | Efficiency $\uparrow$ | Smoothness $\uparrow$ |
> | - | - |-|
> | AD-MLP | 48.45 | 22.63 |
> | TCP-traj  | 76.54 | 18.06 |
> | TCP-ctrl  | 55.97 | 51.91 |
> | UniAD-Base | 129.21 | 43.58 |
>
> We could observe that AD-MLP has the lowest efficiency due to its quick failure and stuck. UniAD has higher efficiency and smoother trajectories compared to TCP-traj, demonstrating the effectivenes of the UniAD's post optimization for the planning head.

---

> > ### Author Response · Authors · 2024-08-19
> >
> > > **Q2: The discussion of potential limitations and challenges associated with the proposed benchmark, such as the gaps between simulation and real-world driving, should be expanded.  The benchmark is currently limited to the CARLA simulation environment, which may not fully capture the complexities of real-world driving scenarios. The reliance on simulated data might introduce biases that are not present in real-world driving, necessitating careful interpretation of the benchmark results. Future work could explore integrating real-world data to enhance the realism of the evaluation.**
> >
> >  We will modify our limitation section to better position this work and provide more information regarding the drawback of the benchmark to its users and discuss the future work in the community. Please check the following text and give any addtional points to improve it. Thanks!
> >
> > *Since the rendering of simulation in CARLA has large gaps compared to real world, utilizing real world datasets could be complementary as done in the concurrent work - NAVSIM[I]. Actually, there is a dilemma in the field for the evaluation of end-to-end autonomous driving algorithms:*
> >
> > | Method | Pros | Cons |
> > | - | - |-|
> > | Real World Datasets | Realistic Rendering | Non-reactive Rendering |
> > | Simulation | Reactive Rendering | Game Rendering |
> >
> > *Diffusion generation might have the potential to provide realistic and reactive rendering, with some pioneering works in the field. However, the illusion and artifact issue of diffusion requires further exploration.*
> >
> > [I] NAVSIM: Data-Driven Non-Reactive Autonomous Vehicle Simulation and Benchmarking. arXiv 2406.15349.
> >
> >
> > > **Q3: a more in-depth analysis of the computational complexity and resource requirements for running the benchmark**
> >
> > Thanks. We will provide those numbers in the updated manuscript:
> >
> > Train TCP & AD-MLP on base set: 1 A6000 GPU around 1 day.
> > Evaluate TCP & AD-MLP: 4 A6000 * 6 hours.
> > Train UniAD on the base set: 8 * A100/H800 - BEVFormer (3 days), Stage 1 (3 days), Stage 2 (3 days)
> > Train VAD on the base set: 8 * A100/H800 - 5 days.
> > Evaluate UniAD/VAD: 8 * A100/H800 - around 2 days.
> >
> > Note that the evaluation process could be linearly speed up or slow down with more or less GPUs since more GPUs only result in more routes to be run in parallel.
> >
> > > **Q4: Providing more detailed comparisons with existing benchmarks and highlighting the unique advantages of Bench2Drive.**
> >
> > We provide comparisions with existing benchmarks in Table 1 of the manuscript and please provide any new point and we are glad to make the comparisons more comprehensive.
> >
> > As for the unique advantages, we will add in the caption of Table 1 that: *Bench2Drive is the only benchmark to evaluate the E2E-AD methods under closed-loop with multi-ability analysis.*
> >
> > > **Q5: The quality of existing figures should be further enhanced.**
> >
> > Thanks for your careful reading and kind advice!
> >
> > We will add in the Fig. 2 and Fig. 4's captions that *Different color represents different routes. Compared to CARLA Leaderboard V2, Bench2Drive has shorter and more diverse routes.*
> >
> > We will adjust the font size of Fig. 3 and Fig. 6.
> >
> > Following you and reviewer WLvp's kind advice, we will move Figure 5 to the appendix and add more explainations in the text.
> >
> >
> > > **Q6: A more thorough discussion of the potential societal impacts and safety considerations of using Bench2Drive for evaluating AD systems could be included to address ethical concerns comprehensively.**
> >
> > We will update the social impact section and please check the following text and please give any addtional points to improve it. Thanks!
> >
> > *Social and Ethical Impact:
> > Social Imapact: The deployment of AD systems holds immense potential to revolutionize transportation, but it also brings significant ethical and safety concerns. Bench2Drive could serve as a platform for rigorously validating the capabilities of AD systems in a controlled and simulated environment, helping to identify potential flaws before real-world deployment.*
> >
> > *Risks of Simulation-Based Evaluation: One of the primary risks is the simulation-reality gap—the difference between how an AD system performs in simulation versus in the real world. Simulations have the difficulties to fully replicate the complexities and unpredictability of real-world driving conditions. There is a risk that an AD system might perform well in simulation but fail in real-world scenarios due to unmodeled factors like rare edge cases, unexpected human behaviors, or varying environmental conditions. Bench2Drive is intended to complement, not replace, real-world testing, and it is crucial to emphasize that simulation is one part of a broader validation process that must include extensive on-road testing.*

---

> > > ### Comment · Reviewer_K6WS · 2024-08-27
> > >
> > > Thanks to the authors for the detailed rebuttal and additional insights provided. I have read the responses and found them informative, particularly the inclusion of new metrics for driving efficiency and smoothness, as well as the expanded discussion on simulation versus real-world scenarios. These additions help to clarify some of the concerns raised and provide a clearer picture of the benchmark’s capabilities and limitations.
> > >
> > > After considering the rebuttal and feedback from other reviewers, I will maintain my current rating. I encourage the authors to incorporate the suggested revisions and improvements into the manuscript to enhance its quality and impact.

---

> > > > ### Author Response · Authors · 2024-08-27
> > > >
> > > > Thanks for your reply and advice!  Sure, we will update the manuscript with the revisions.

---

### Official Review · Reviewer_XSJo · 2024-07-23
**Bench2Drive Paper Review**

**Rating:** 6
**Confidence:** 4
**Correctness:** The evaluation methods seems to be ap…
**Clarity:** The paper could improve its legibility

**Review:**

End-to-end AD systems are rapidly becoming more and more common, and establishing benchmarks is of utmost importance to make sure we are comparing apples to apples.
The paper has some interesting ideas which I think should have been executed a bit better:
* Establishing a fixed training set is a good idea, but since this is simulated data I would have appreciated more details about the data generation procedure and the expert model choice.
* The skills definition is interesting but it's lacking details: how were the scenarios for each subset selected? Was this done programmatically (if yes, can you share the code) or manually? Or is there only one scenario per class?

Some parts of the paper could be formatted better for legibility, and there are a bunch of typos scattered around.

**Strengths:**

* The scenario-based evaluation can give more signal that the complex CARLA test.
* Providing a fixed dataset can help regularize benchmarks.
* Evaluated several sota methods in their benchmark

**Additional Feedback:**

No additional feedback

**Documentation:**

The benchmark documentation is provided on github

**Ethics:**

No ethical concerns, as it's simulated data

**Limitations:**

I think this limitation section is missing a bit more context of this sim-to-real gap. By using a real-world dataset directly, we would have access to both the ground truth sensor data and expert (defined as the human operator this time). So I think the authors should highlight how this benchmark could be complemented with a non-simulated data-based benchmark (likely in open-loop).

**Opportunities For Improvement:**

* Add a more detailed explanation of why ThinkDrive was chosen as the data generation model.
* When comparing feature distillation, the comparison of the same method (e.g. TCP) with and without expert feature distillation is missing, so differences are conflated between the method itself and having access to expert features.
* General formatting: e.g. fix typos below, Figure 5 could definitely be in Appendix and leave more room to explain other concepts better.


[Typo nits]
Line 51: shown
Line 60: expert
Line 89: closed

**Relation To Prior Work:**

Prior work is sufficiently discussed

**Summary And Contributions:**

This paper introduces the Bench2Drive benchmark, a closed-loop benchmark based on the CARLA simulator for E2E AD systems.
The paper makes the following contributions:
* Releases a large scale training set for comparability between benchmarks.
* Evaluates several state-of-the-art methods on the new benchmark
* Introduces "multi-ability" subsets for specific type of maneuvers

---

> ### Author Rebuttal · Authors · 2024-08-19
>
> Thanks for your acknowledgement and kind advice. Regarding your concerns, we give responses below:
>
> > **Q1: Add a more detailed explanation of why Think2Drive was chosen as the data generation model.**
>
> To collect data in the simulation, it either requires human to drive in the simulation which is expensive or uses an expert model with privileged information to drive in the simulation. During the dataset curation process, Think2Drive is the only model in the community being able to solve all scenarios.
>
> We will add the explanation in the manuscript.
>
> > **Q2: The comparison of the same method (e.g. TCP) with and without expert feature distillation is missing**
>
> Good point! We train TCPs with and without expert distillation and conduct the closed-loop evalution. Note that evaluation protocal is not exactly the same as in the paper since (I) we introduce the new metric about efficiency as suggested by reviewer WLvp and thus the punishment of min speed infraction is removed when calculating driving score. We adjust the tickruntime for better robustness. We are re-running all methods and will update the results in the Github. We provide the comparison of the two TCPs below:
>
> | Method | Driving Score $\uparrow$ | Success Rate $\uparrow$ |
> | - | - |-|
> | TCP-traj with expert distillation  | 59.57 | 73.18 |
> | TCP-traj  without expert distillation | 49.30 | 60.91 |
>
> We could observe a significant performance drop when removing expert distillation, aligning with findings in the community.
>
>
> > **Q3: Typos & Figure 5**
>
> Yes, they are typos and thanks for your careful reading! Following you and reviewer K6WS's kind advice, we will move Figure 5 to the appendix and add more explainations in the text.
>
> > **Q4:  I think the authors should highlight how this benchmark could be complemented with a non-simulated data-based benchmark**
>
> Agreed. We will modify our limitation section. Please check the following text and give any addtional points to improve it.
>
> *Since the rendering of simulation in CARLA has large gaps compared to real world, utilizing real world datasets could be complementary as done in the concurrent work - NAVSIM[I]. Actually, there is a dilemma in the field for the evaluation of end-to-end autonomous driving algorithms:*
>
> | Method | Pros | Cons |
> | - | - |-|
> | Real World Datasets | Realistic Rendering | Non-reactive Rendering |
> | Simulation | Reactive Rendering | Game Rendering |
>
> *Diffusion generation might have the potential to provide realistic and reactive rendering, with some pioneering works in the field. However, the illusion and artifact issue of diffusion requires further exploration.*
>
> [I] NAVSIM: Data-Driven Non-Reactive Autonomous Vehicle Simulation and Benchmarking. arXiv 2406.15349.

---

> > ### Comment · Reviewer_XSJo · 2024-08-27
> >
> > > To collect data in the simulation, it either requires human to drive in the simulation which is expensive or uses an expert model with privileged information to drive in the simulation. During the dataset curation process, Think2Drive is the only model in the community being able to solve all scenarios.
> >
> > The necessity of an expert model is clear, what is not clear is:
> > 1. what could have been the possible choices here (e.g. uniAD?)
> > 2. If this model (and weights) are open-sourced
> > 3. What scenarios are the authors referring to here? Can the authors report a reference to these?
> >
> > Since the expert policy is a central component for the proposed dataset, I think a more extensive discussion around that choice is needed.
> >
> > The other answers to Q2, Q3 and Q4 have addressed the rest of my issues, thanks for reviewing!

---

> > ### Author Response · Authors · 2024-08-27
> >
> > Thanks for you reply!  We are glad to discuss regarding this topic:
> >
> > > **What could have been the possible choices here (e.g. UniAD?)**
> >
> > The expert is responsible for collecting the data so that **student models could learn from the data**. In the real world, this is usually done by human to drive around the city, like the curation of KITTI, nuScenes, Waymo, Argoverse.  However, it requires lots of human efforts.  **In simulation, there is a good&cheap substitute - teacher model.** The teacher model would use information not available in the real world (termed *privileged information* in the community), for example, ground-truth localtions, states, and intentions of surrounding agents and ground-truth states of traffic lights, etc.  As a result, people using CARLA either write rules[1] or train a RL model[2,3] to use the privileged information to drive in the simulation.  Thus, UniAD is not feasible for this task, as it is an imitation learning model, i.e., supervised learning of expert planning trajectories. Actually,  models like UniAD is called student model [1,2,3,4] since it requires teacher model to collect data to *teach* it.
> >
> > > **If this model (and weights) are open-sourced**
> >
> >  Think2Drive is not open sourced. However, we have discussed with the authors of recently released PDM-Lite [5] and their rule-based model is able to run in CARLA Leaderboard 2.0 as well. Actually, from our discussions with users in the issue section of Bench2Drive, there are already users successfully adopting PDM-Lite to collect their own data.
> >
> > However, we would like to point out that one important contributions of Bench2Drive is to provide a fixed yet comprehensive enough datasets so that the comparison is fair. Otherwise, as pointing out in [6,7], the influence of data is  huge and thus the comparison is less sound.
> >
> > > **What scenarios are the authors referring to here? Can the authors report a reference to these?**
> >
> > They are challenging driving events initially created by CARLA and enriched by our team.  You may refer to CARLA's official docs: https://leaderboard.carla.org/scenarios/.   In Appendix D of the submission, we give descriptions as well. For the code implementations, you may refer to our repo: https://github.com/Thinklab-SJTU/Bench2Drive/tree/main/scenario_runner/srunner/scenarios.
> >
> > **We totally agree that the expert and dataset curation part should be extended in the manuscript and will add the aforementioned information in the camera-ready version.** Thanks for your advice!
> >
> > [1] TransFuser: Imitation with Transformer-Based Sensor Fusion for Autonomous Driving. IEEE TPAMI
> >
> > [2] End-to-End Urban Driving by Imitating a Reinforcement Learning Coach. ICCV 2021
> >
> > [3] Think2Drive: Efficient Reinforcement Learning by Thinking in Latent World Model for Quasi-Realistic Autonomous Driving (in CARLA-v2). ECCV 2024
> >
> > [4] Coaching a Teachable Student. CVPR 2024
> >
> > [5] https://github.com/autonomousvision/carla_garage/tree/leaderboard_2
> >
> > [6] PlanT: Explainable Planning Transformers via Object-Level Representations. CoRL 2022
> >
> > [7] Hidden Biases of End-to-End Driving Models. ICCV 2023

---

> > > ### Comment · Reviewer_XSJo · 2024-08-27
> > >
> > > Thank you for the in-depth response. If the authors can integrate this information in the camera-ready version, I'm happy to update my vote to 6.

---

> > > > ### Author Response · Authors · 2024-08-27
> > > >
> > > > Sure, we will integrate them. Thanks for your time and advice!

---

### Official Review · Reviewer_WLvp · 2024-07-26

**Rating:** 6
**Confidence:** 4
**Clarity:** The overall presentation of this pape…

**Review:**

This paper proposes a new dataset for closed-loop E2E-AD training and evaluation. The overall presentation is good and the proposed dataset consists of large scale clips and diverse scenarios. Detailed pros and cons please refer to the following section.

**Strengths:**

1. Large Scale: Collecting a large scale dataset including 2 million fully annotated frames with nuScenes like sensor configuration, sourced from 10000 clips.
2. Diversity: Including different diverse scenarios such as cut-ins, overtakings under different weather conditions and geometric locations.
3. Closed-Loop Multi-Ability Evaluation Protocol: Design 5 short routes for 44 scenarios to evaluate granular performance.
4. Comprehensive Baselines: Proving several recent E2E-AD algorithms implementation and conduct experiments on the proposed dataset.

**Additional Feedback:**

1. Providing visualization videos between different baseline methods on different scenarios will make reader easy to explore the dataset and the performances of E2E-AD methods on the proposed dataset.

**Correctness:**

The dataset is constructed in a sound way. The author conduct a large scale and diverse dataset with different scenarios.

**Documentation:**

The authors release all data, codes and checkpoints in GitHub and huggingfaces.

**Ethics:**

There are not ethical concerns with the submission.

**Limitations:**

The author discuss the limitation of the proposed work and propose to adopt generative models for realistic rendering in the future work.

**Opportunities For Improvement:**

1. The limitation of evaluation metric. The current metric focus more on the goal achieving and lacks the measurement of driving efficiency and smoothness of driving trajectories.
2. Missing descriptions about the policy of agents during the driving simulation.

**Relation To Prior Work:**

The author discuss the relationship between the proposed dataset and previous works. Table 1 shows the comparison between the proposed dataset and other planning related datasets.

**Summary And Contributions:**

This paper focuses on the evaluation issue of closed-loop end-to-end autonomous driving (E2E-AD). Recent open-loop pipelines can not reflect the real performance of E2E-AD algorithms and the current closed-loop benchmarks can not ensure fair apple-to-apple comparison. To address the issue, this paper proposed Bench2Drive, a benchmark for evaluating E2E-AD systems in a closed-loop manner.

---

> ### Author Rebuttal · Authors · 2024-08-19
>
> Thanks for your acknowledgement and kind advice. Regarding your concerns, we give responses below:
>
>
> > **Q1: Lacks the measurement of driving efficiency and smoothness of driving trajectories.**
>
> Thanks for your kind advice! We implement two additional metrics to evaluate the two perspectives:
>
> - **Efficiency**: The CARLA team has implemented a function to check whether the self-driving car's speed is too low. This is determined by comparing the vehicle's speed with nearby vehicle:
> $$\text{Speed Percentage}=\frac{\text{Ego Vehicle's Speed}}{\text{Average Speed of Nearby Vehicles}} $$
> This function calculates the speed percentage using the vehicle’s speed and the average speed of nearby vehicles at current frame. CARLA Leaderboard sets four checkpoints per route and checks the ego vehicle's speed when the ego vehicle arrives a checkpoint. Specifically, if the vehicle is faster than nearby vehicles, the driving efficiency would be larger than 100%. The check results are included as a penalty in the final driving score. However, with only four checkpoints, the vehicle must cover 25% of the total route distance before reaching the next checkpoint. This leads to a high variance in the penalty values for low speeds, complicating the reflection of driving capabilities in the driving scores. To alleviate this, we increase the number of checkpoints to 20. Speed check is now performed every 5% of the total route length, and it is excluded from the driving score calculation. The final driving efficiency metric is defined as the average of the speed percentage over all checks.
> $$\text{Driving Efficiency}=\frac{\sum_i \text{Speed Percentage}\phantom{}_i}{\text{Speed Check Times}} $$
> If the ego vehicle fails to pass the initial 5% checkpoint, this route is not included in the final driving efficiency metric calculation. To account for cases where abnormal speed spikes may occur (e.g., when the vehicle falls off the current map layer), speed percentage values exceeding 1000% are filtered out.
>
> - **Smoothness**: Smoothness is closely related to human experience and thus requires comparing autonomous driving policy with the behavior of numerous human driving experts to measure it. For this, we follow the popular benchmark nuPlan's smoothness(also called comfort) protocol, which evaluates ego’s minimum and maximum longitudinal accelerations, the maximum absolute values of lateral acceleration, yaw rate, yaw acceleration, the longitudinal component of jerk, and the maximum magnitude of the jerk vector. These variables are compared to thresholds with default values determined empirically from the examination of nuPlan’s human expert trajectories. Smoothness is measured based on whether these values fall within the upper and lower bounds of the expert values.
> $$\small \text{Frame Variable Smoothness(FVS)} = \begin{cases}
> \text{True} & \text{if lower bound}\leq p_{\,i} \leq \text{upper bound}, \\
> \text{False} & \text{otherwise}
> \end{cases} \\
>  p \in \text{smoothness vars}, \, 0\leq i \leq \text{total frames}$$
>
>     - smoothness variables(vars) include:
>       - `longitudinal_acceleration, expert bound:[-4.05, 2.40]`
>       - `maximum_absolute_lateral_acceleration, expert bound:[-4.89, 4.89]`
>       - `yaw_rate, expert bound:[-0.95, 0.95]`
>       - `yaw_acceleration, expert bound:[-1.93, 1.93]`
>       - `longitudinal_component_of_jerk, expert bound:[-4.13, 4.13]`
>       - `maximum_magnitude_of_jerk_vector, expert bound:[-8.37, 8.37]`
>
> A trajectory is deemed Smooth only if all smoothness variables meet the smoothness criteria.
> $$\text{Trajectory Smoothness} =\bigwedge_{i=0}^{\text{total frames}}\text{FVS}
> $$
> In nuPlan, smoothness is determined by frame-by-frame evaluation of these variables over the entire trajectory, which makes it susceptible to local driving behaviors. For example, if a vehicle ahead suddenly brakes, the ego vehicle must also brake abruptly to avoid a collision. Even if the ego’s hard brake behavior is appropriate in this case and its driving is smooth at other times, the entire trajectory could still be judged as unsmooth, leading to unreasonable evaluation results. To mitigate this issue, we segment the entire trajectory at a timestep interval $\small n=20$ for evaluation.
> $$
> \text{Segment Smoothness} =\bigwedge_{i=\text{start frame}}^{\text{end frame}}\text{FVS}
> $$
> The final smoothness metric is defined as the ratio of smooth trajectory segments to the total number of segments.
> $$
> \text{Smoothness}=\frac{\text{Number of Smoothness Segments}}{\text{Total Segments}}
> $$
> Specifically, if the ego vehicle is blocked (speed remains below 0.1 for more than 60 seconds.), resulting in a failure case, this segment is still be considered as smooth because its speed is safe for human. Note that if the total frames of a trajectory are less than 20, the respective route is excluded from the smoothness assessment.
>
>
> We are re-runing all baselines to get the two new metrics while some of them are not finished yet due to the heavy computational buderns and we will update the full results in the Github. The following are results of finished ones:
>
> | Method | Efficiency $\uparrow$ | Smoothness $\uparrow$ |
> | - | - |-|
> | AD-MLP | 48.45 | 22.63 |
> | TCP-traj  | 76.54 | 18.06 |
> | TCP-ctrl  | 55.97 | 51.91 |
> | UniAD-Base | 129.21 | 43.58 |
>
> We could observe that AD-MLP has the lowest efficiency due to its quick failure and stuck. UniAD has higher efficiency and smoother trajectories compared to TCP-traj, demonstrating the effectivenes of the UniAD's post optimization for the planning head.

---

> > ### Author Response · Authors · 2024-08-19
> >
> > > **Q2: Descriptions about the policy of agents during the driving simulation.**
> >
> > All baselines models either: (I) generate actions, which is throttle (0-1), brake (0-1), steer in (-1-+1) in CARLA and we directly feed the action to control the vehicle (II) generate plans of the ego agent's future trajectories and we follow the widely used two PID strategies: one for longitudinal control (throttle and brake) and one for lateral (steer). We feed the output of PIDs to control the vehicle. We will add those information into the manuscript.
> >
> >
> > > **Q3: Provide visualization videos between different baseline methods on different scenarios.**
> >
> > Thanks for your kind advice. We conduct visualizations and upload results to Bench2Drive's Github: https://github.com/Thinklab-SJTU/Bench2DriveZoo/blob/uniad/vad/analysis/analysis.md.  For all five abilities, we choose some representative scenarios to visualize, where some baselines success and some baselines fail for the ease of comparison and analysis. We give the corresponding failure analysis so that the users and practioners could have a sense about the pros, cons, and future works of existing E2E-AD methods.

---

> > > ### Comment · Reviewer_WLvp · 2024-08-27
> > >
> > > Thanks for your response.
> > > The responses for Q1 and Q3 have addressed my concerns. However, for Q2, the author should additionally provide the policy for NPC agents. For readers unfamiliar with Carla, it's unclear how NPC behavior is defined and whether they interact with the EGO vehicle.

---

> > > > ### Author Response · Authors · 2024-08-27
> > > >
> > > > Thanks for your reply!  **We provide the detail about NPC agents below**:
> > > >
> > > > In CARLA, three behavior types are preset in [CARLA Agent](https://carla.readthedocs.io/en/latest/adv_agents/#behavior-types): **cautious**, **normal**, and **aggressive**. These behavior types govern the driving actions of NPCs, influencing factors such as speed, responses to other vehicles, and safety protocols. The key parameters for each behavior mode include:
> > > >
> > > > - **max_speed**: Sets the maximum speed (km/h) that an NPC vehicle can reach.
> > > > - **speed_lim_dist**: Value in km/h that defines how far your vehicle's target speed will be from the current speed limit.
> > > > - **speed_decrease**: Controls the deceleration of the NPC when approaching a slower vehicle ahead.
> > > > - **safety_time**: Estimates the time to a collision if the vehicle in front suddenly brakes.
> > > > - **min_proximity_threshold**: Defines the minimum distance before the NPC takes actions such as evasive maneuvers or tailgating.
> > > > - **braking_distance**: The distance at which the NPC performs an emergency stop to avoid a collision.
> > > > - **tailgate_counter**: A counter that prevents the NPC from initiating a new tailgating action too soon after the last one.
> > > >
> > > > These behavior designs interact with the ego (self-driving) vehicle, ensuring that NPCs respond to the presence and actions of the ego vehicle in the simulation. The parameters for different behavior styles are as follows:
> > > >
> > > > | **Parameter** | **Cautious** | **Normal** | **Aggressive** |
> > > > |---------------|--------------|------------|----------------|
> > > > | **max_speed (km/h)** | 40 | 50 | 70 |
> > > > | **speed_lim_dist (km/h)** | 6 | 3 | 1 |
> > > > | **speed_decrease (km/h)** | 12 | 10 | 8 |
> > > > | **safety_time (seconds)** | 3 | 3 | 3 |
> > > > | **min_proximity_threshold (meters)** | 12 | 10 | 8 |
> > > > | **braking_distance (meters)** | 6 | 5 | 4 |
> > > > | **tailgate_counter (times)** | 0 | 0 | -1 |
> > > >
> > > > *Table: Comparison of Cautious, Normal, and Aggressive behavior parameters.*
> > > >
> > > > The rule-based behavior decision algorithm in [behavior_agent.py](https://github.com/carla-simulator/carla/blob/dev/PythonAPI/carla/agents/navigation/behavior_agent.py) for non-player character (NPC) vehicles is shown below:
> > > >
> > > > Algorithm: Non-Player Character (NPC) Vehicles Behavior Decision
> > > > 1. **Update vehicle’s surrounding information**
> > > > 2. **if** red light **or** stop sign detected:
> > > >     - **return** `emergency_stop()`
> > > > 3. **if** pedestrian detected **and** within braking distance:
> > > >     - **return** `emergency_stop()`
> > > > 4. **if** other vehicle nearby:
> > > >     - **if** within braking distance:
> > > >         - **return** `emergency_stop()`
> > > >     - **else**:
> > > >         - **return** `car_following()`
> > > > 5. **else if** ego vehicle at intersection **and** (turn left or turn right):
> > > >     - `adjust_speed_limit()`
> > > >     - **return** `PID_Control`
> > > > 6. **else if** in normal driving:
> > > >     - `maintain_speed_limit()`
> > > >     - **return** `PID_Control`
> > > >
> > > > The behavior of walkers/pedestrians is characterized by their consistent adherence to the route line at a constant speed, with the inability to walk backward. In autonomous driving scenarios, pedestrian safety is of utmost importance. Pedestrians have the highest priority when autonomous vehicles interact with humans, and autonomous vehicles should learn to give way to pedestrians regardless of traffic conditions.
> > > >
> > > >
> > > > **We will update those information  in the camera ready version of the paper.  Please let us know if there are any unclear part.**

---

> > > > > ### Author Response · Authors · 2024-08-30
> > > > >
> > > > > Dear Reviewer,
> > > > >
> > > > > As the discussion stage will end in two days, please tell us if there is any further concern and we are glad to discuss. Respectfully, we request that you consider improving the score if you are satisfying with the major update of Bench2Drive. Thanks!

---

### Official Review · Reviewer_9tDx · 2024-07-28

**Rating:** 8
**Confidence:** 4
**Correctness:** Yes
**Clarity:** Yes

**Review:**

Strengths:
1. This paper is well-written, with a clear motivation addressing a longstanding concern in the field.
2. The contributions regarding datasets and benchmarks are substantial, significantly advancing the development of closed-loop autonomous driving.
3. It is the first to evaluate the performance of existing open-loop and closed-loop methods on the same benchmark, providing valuable insights.

Weaknesses:
1. Can the proposed multi-ability evaluation be applied to real-world datasets?
2. It would be beneficial to provide a failure case study of existing methods on the new benchmark.

**Strengths:**

Strengths:
1. This paper is well-written, with a clear motivation addressing a longstanding concern in the field.
2. The contributions regarding datasets and benchmarks are substantial, significantly advancing the development of closed-loop autonomous driving.
3. It is the first to evaluate the performance of existing open-loop and closed-loop methods on the same benchmark, providing valuable insights.

**Additional Feedback:**

See the weaknesses.

**Documentation:**

Yes

**Limitations:**

Yes

**Opportunities For Improvement:**

Weaknesses:
1. Can the proposed multi-ability evaluation be applied to real-world datasets?
2. It would be beneficial to provide a failure case study of existing methods on the new benchmark.

**Relation To Prior Work:**

Yes

**Summary And Contributions:**

This paper present a new benchmark named Bench2Drive for closed-loop evaluation of end-to-end autonomous driving methods. A fully-annotated large-scale dataset and a multi-ability evaluation toolkit are open sourced for driving skill assessment. SOTA end-to-end driving methods are tested on the proposed benchmark with more evaluation metrics.

---

> ### Author Rebuttal · Authors · 2024-08-19
>
> Thanks for your acknowledgement and kind advice. Regarding your concerns, we give responses below:
>
> > **Q1: Can the proposed multi-ability evaluation be applied to real-world datasets?**
>
> In principle, the real world datasets could be split into short clips and then manually classified into different behaviors to conduct multi-ability evaluation.
>
> However, the biggest issue of evaluation with real world datasets is that when the ego agent conduct behaviors different from recorded ones, it is hard to provide the corresponding sensor information (for exmaple, images). As a result, the concurrent work - NAVSIM (NA stands for Non-Reactive)[I] assumes that the sensor information does not change, regardless of what the ego vehicle has done.
>
> In summary, there is a dilemma in the field for the evaluation of end-to-end autonomous driving algorithms:
>
> | Method | Pros | Cons |
> | - | - |-|
> | Real World Datasets | Realistic Rendering | Non-reactive Rendering |
> | Simulation | Reactive Rendering | Game Rendering |
>
> As mentioned in the limitation section of the manuscript, diffusion generation might have the potential to provide realistic and reactive rendering, with some pioneering works in the field. However, the illusion and artifact issue of diffusion requires further exploration.
>
> [I] NAVSIM: Data-Driven Non-Reactive Autonomous Vehicle Simulation and Benchmarking. arXiv 2406.15349.
>
> We will add this discussion to the manuscript to better position this work and provide more information regarding the drawback of the benchmark to its users and discuss the future work in the community.
>
> > **Q2: Failure case study of existing methods on the new benchmark.**
>
> Thanks for your kind advice. We conduct visualizations and upload results to Bench2Drive's Github: https://github.com/Thinklab-SJTU/Bench2DriveZoo/blob/uniad/vad/analysis/analysis.md.  For all five abilities, we choose some representative scenarios to visualize, where some baselines success and some baselines fail for the ease of comparison and analysis. We give the corresponding failure analysis so that the users and practioners could have a sense about the pros, cons, and future works of existing E2E-AD methods.

---

> > ### Comment · Reviewer_9tDx · 2024-08-27
> >
> > Thanks a lot for the detailed feedback, which solved my main concerns. I strongly suggest that the authors include the corresponding experiments and discussions of the rebuttal part in camera ready version.

---

> > > ### Author Response · Authors · 2024-08-27
> > >
> > > Thanks for your reply!  Sure, all updates have been pushed to the Github repo and we will update the paper accordingly.

---

### Author Rebuttal · Authors · 2024-08-19

We express our gratitude to all reviewers for their valuable time and insightful comments. The acknowledgement about the closed-loop multi-ability evaluation (9tDx, WLvp, XSJo, K6WS), comprehensive baselines (9tDx, WLvp, XSJo, K6WS), large-scale and diverse fixed dataset (9tDx, WLvp, XSJo, K6WS) is encouraging.

Additionally, we are glad to share that: since the release in May, 2024, Bench2Drive has been used by lots of members in the community, evidenced by 800+ stars, 62+28=90 issues, and numerous emails to consult about details and collaborations. The authors of state-of-the-art end-to-ned autonomous driving works like SparseAD, SparseDrive, Hydra-MDP are transferring their methods into Bench2Drive. The community discussions also help us to improve the documentations and codes about visualizations and the evaluation process. We are thrilled by the wide usage within the community and will continuously maintain the benchmark.

Following your kind advice, in the updated version of Bench2Drive Github Repo, we:
- **Add two new metrics regarding the efficiency and smoothness of the driving, beyond the goal achieving ability**. The code has been pushed to Github and we are re-evaluating all baselines with the two metrics and will update all results to Github.
- **Add visualization videos of baseline methods for failure case analysis** to give users and practioners a better sense about the pros, cons, and future works for end-to-end autonomous driving methods. The link is at https://github.com/Thinklab-SJTU/Bench2DriveZoo/blob/uniad/vad/analysis/analysis.md
- **Add discussions regarding the sim-to-real gaps**, complementary relation with real-world non-reactive evaluation, and the potential solution - AIGC technology.
- **Fix typos and figures**

---

### Author Response · Authors · 2024-08-26
**Humble request for feedbacks**

Dear AC and Reviewers,

As the author-reviewer discussion phase nears its end, we hope our responses have addressed all your concerns. If there are any further questions or issues, we are eager to discuss them.

Though Bench2Drive's closed-loop multi-ability evaluation, comprehensive baselines, and the extensive dataset have been recognized by all reviewers,  we feel that the current ratings are conservative, especially given the high standards of the NeurIPS Dataset Track.

**During the rebuttal period, we have made significant updates to baselines, metrics, and visualizations after weeks of efforts and lots of computational power. We respectfully request that you consider improving the score if these updates have resolved your concerns**.

Bench2Drive is gaining traction in the community, with 892 stars, 64 forks, 106 issues (99 resolved) as of today, which we believe is a valuable contribution to the NeurIPS Datasets and Benchmarks Track and worth sharing to the community.

Looking forward to your feedback.

Sincerely,

Authors of Submission 114

---

### Decision · Program_Chairs · 2024-09-26

**Decision:**

Accept (Poster)

**Comment:**

The reviewers and I are in agreement that this paper is ready for acceptance, as is advances the benchmarks for closed-loop autonomous driving. There was a significant amount of discussion around the limitations such as the reliance on only simulated data from CARLA and potential sim-to-real gaps here, but overall these have been resolved. We encourage the authors to continue incorporating any other points from these discussion into the final version of the paper and in the continued maintenance of the benchmark for the community.